# Codoped porous carbon nanofibres as a potassium metal host for nonaqueous K-ion batteries

Siwu Li [1], Haolin Zhu[1], Yuan Liu[2], Zhilong Han[1], Linfeng Peng[1], Shuping Li[1], Chuang Yu[1], Shijie Cheng[1] & Jia Xie [1] ✉

Potassium metal is an appealing alternative to lithium as an alkali metal anode for future electrochemical energy storage systems. However, the use of potassium metal is hindered by the growth of unfavourable deposition (e.g., dendrites) and volume changes upon cycling. To circumvent these issues, we propose the synthesis and application of nitrogen and zinc codoped porous carbon nanofibres that act as potassium metal hosts. This carbonaceous porous material enables rapid potassium infusion (e.g., < 1 s cm$^{-2}$) with a high potassium content (e.g., 97 wt. %) and low potassium nucleation overpotential (e.g., 15 mV at 0.5 mA cm$^{-2}$). Experimental and theoretical measurements and analyses demonstrate that the carbon nanofibres induce uniform potassium deposition within its porous network and facilitate a dendrite-free morphology during asymmetric and symmetric cell cycling. Interestingly, when the potassium-infused carbon material is tested as an active negative electrode material in combination with a sulfur-based positive electrode and a nonaqueous electrolyte solution in the coin cell configuration, an average discharge voltage of approximately 1.6 V and a discharge capacity of approximately 470 mA h g$^{-1}$ after 600 cycles at 500 mA g$^{-1}$ and 25 °C are achieved.

With the rapid advancement of portable electronics, electrified transportation and smart grids, rechargeable batteries, especially lithium-ion batteries (LIBs), are facing huge consumption demands, which is causing concern about limited resources[1–3]. Consequently, alternative battery systems using earth-abundant elements are of great interest to scientists and industries. Potassium-ion batteries, for instance, are drawing extensive attention for their comparable operation voltages and power density with regard to those of LIBs[4,5]. More importantly, K is nearly inexhaustible because of its high abundance, which is three orders of magnitude greater than that of Li in the Earth's crust (Li: 0.0017 wt. %, K: 1.5 wt. %)[6]. These advantages endow K with considerable potential for large-scale energy storage applications.

To date, several types of anode materials have been developed to support aprotic K batteries, such as intercalation type (graphite, hard/soft carbon, Ti oxides, MoO$_2$, Nb$_2$O$_5$, VS$_2$)[7–12], conversion type (FeVO$_4$, FeS$_2$, SnS$_2$, Sb$_2$S$_3$, MoS$_2$)[13–17], alloy type (Sn, Sb, Bi, P)[18–21], organic type (carboxylates, metal-organic frameworks or MOFs, covalent-organic frameworks or COFs)[22–24], and other types (MXenes and polyanionic compounds)[25,26]. However, to achieve high specific energy in K batteries, K metal is an optimal choice based on its lowest redox potential (−2.93 V vs. SHE) and highest specific capacity (687 mA h g$^{-1}$) among all the anode materials[27–29]. Beyond that, the participation of K metal can also realize the application of K-free cathodes to fabricate high-specific-energy K battery systems, such as K−S (1023 Wh kg$^{-1}$) and K−O$_2$ (935 Wh kg$^{-1}$) batteries[30,31].

[1]State Key Laboratory of Advanced Electromagnetic Engineering and Technology, School of Electrical and Electronic Engineering, Huazhong University of Science and Technology, Wuhan 430074, China. [2]Key Laboratory for Renewable Energy, Beijing Key Laboratory for New Energy Materials and Devices, Beijing National Laboratory for Condensed Matter Physics, Institute of Physics, Chinese Academy of Sciences, Beijing 100190, China. ✉e-mail: xiejia@hust.edu.cn

Unfortunately, K metal anodes also encounter inevitable dendrite growth issues during electrochemical plating/stripping, similar to Li and Na, which severely hinders their practical use in K batteries[32]. Specifically, there are several problems that K dendrites can bring about. First, exposed K dendrites with large surfaces can continuously react with the electrolyte, causing low Coulombic efficiency (CE)[29,33]. In addition, the stripping process tends to take place at the roots of the dendrites, which ultimately isolates the dendrites electronically and forms "dead K"[34]. Moreover, the accumulated dendrites can reach the counter electrode, triggering internal short circuits, battery failure, and even safety issues, including fire and explosion[35]. In general, the principal contributor of K dendrites is the unstable solid electrolyte interphase (SEI)[36]. This nonself-passivating SEI can be mechanically damaged during K plating and stripping because of the volume change[37]. As a result of "tip effects", an inhomogeneous electric field distribution and nonuniform $K^+$ flux can be induced on the defective SEI, leading to irregular K deposition and subsequent dendrite growth[38]. Accordingly, researchers have exploited some strategies to alleviate or suppress dendrite growth. For example, Goodenough et al. first alloyed K with Na to form a liquid Na-K anode[39]. This liquid alloy exhibits deformability and self-healing properties, which is favorable for a dendrite-free anode[39,40]. However, the fluidity makes it easier for the anode to cause safety risks under an external force. Specific electrolyte formulations or electrode surface engineering are also effective routes for suppressing K dendrites by tuning the metal surface or SEI[29]. Although an improved interphase can protect the K anode from dendrite formation to a certain extent, it becomes ineffective when the electrode undergoes a large volumetric change during K plating/stripping. Moreover, a conductive host is also considered for stabilizing the K anode. Its advantages are as follows: (1) K metal can be encapsulated in the host and therefore reduce the chance of side reactions between the alkali metal and electrolyte; (2) a host can improve the structural stability of the metal anode by buffering the volume variations; and (3) a host can provide conductive networks that induce fast ion/electron transport and decrease the local current density, further hindering dendrite growth[27]. Typically, metal-based hosts, such as Cu and Al, are a class of potential hosts for stable and dendrite-free composite K anodes considering their commercial availability. After proper treatment to increase the potassiophilicity, these modified hosts, including rGO@3D-Cu (Cu foam coated with reduced graphene oxide), Al@Al (aluminum foil coated with aluminum powders), and $Cu_3Pt$–Cu mesh ($Cu_3Pt$ functionalized-Cu meshes), show improvement in stable K plating and stripping[41–43]. These metal-based hosts all demonstrate improved CE and suppressed overpotential, showing the validity of introducing potassiophilic sites to hosts. However, the high density and relatively low space utilization hinder the application of metal hosts. In contrast, carbon-based hosts are more favorable for K accommodation mainly because of their lightweight and electrochemical stability, which can favor high specific energy and energy density. HNCP/G (hollow N-doped C polyhedrons/graphene composite) and an aligned carbon nanotube membrane (ACM), as pure carbon hosts, can realize a prolonged symmetric cell cycling time (>100 h) and decreased overpotential (<0.1 V)[44,45]. In these cases, robust and electronically conductive carbon-based hosts successfully demonstrate their potential for K accommodation. Consequently, the incorporation of carbon-based materials with other kinds of potassiophilic species has become popular for enhanced K plating/stripping electrochemistry. For example, PM/NiO (puffed millet/NiO) gives an enhanced K deposition volume and smaller voltage hysteresis due to the synergy between the PM host and the potassiophilic NiO nanoparticles[46]. $PCNF@SnO_2$ ($SnO_2$-coated conductive porous carbon nanofibre) realizes high K uptake (87% wt. %@15 mg cm$^{-2}$) and uniform K nucleation on account of its void-rich carbon nanofibres and potassiophilic $SnO_2$ coating[47]. DN-MXene/CNT (defect-rich and nitrogen-containing MXene/carbon nanotube) achieves high CE (98.6%) and prolonged cycle life on account of its titanium defects and interconnected carbon scaffolds[48]. In addition to the above metal compounds, metal nanoparticles might draw attention for their advantages of high K affinity and ease of synthesis. Previously, Au, Ag, and Zn nanoparticles have been proven to be advantageous in leading to the heterogeneous seeded growth of Li on carbon hosts[49–54]. These results suggest that the introduction of metal nanoparticles (Pb, Sb, Sn, and Zn) that are good at forming K alloys would be an efficient strategy for improving the potassiophilicity of carbon hosts.

Altogether, we believe that the following features should be considered when designing a powerful host for high-performance K anodes: (1) high potassiophilicity and chemical/electrochemical stability; (2) good electron and ion transport; (3) lightweight and robust structure; and (4) high space utilization or abundant inner space. Based on these criteria, herein, we demonstrate a highly potassophilic and lightweight freestanding K metal host constructed with porous carbon nanofibres embedded with monodispersed nitrogen sites and zinc clusters. Specifically, the nanosized Zn-triazole metal-organic framework (MOF), MET-6, was introduced to fabricate polyacrylonitrile (PAN)-based pristine nanofibres via an electrospinning method. MET-6-containing fibers decompose and generate large amounts of $N_2$ gases under controlled pyrolysis, which forms hierarchically porous carbon fibers in situ[55,56]. Taking advantage of the nanoconfinement from the coordination bonds in MOF crystals, the derivative possesses monodispersed amorphous Zn clusters without agglomeration. In addition, the nitrogen-rich PAN and MET-6 also provide enough nitrogen sites for the nanofibres. With these merits, the MET-derived carbon nanofibres, denoted as MSCNFs, achieved the following results, which are verified by a series of in situ/ex situ characterizations and computational analyses: (1) the monodispersed Zn and N sites together enable a potassiophilic behavior that maintains low nucleation overpotential (15 mV at 0.5 mA cm$^{-2}$); (2) the hierarchical pores together with the voids in the interconnected nanofibres guarantee the ultrafast infusion (less than 1 s over 1 cm$^2$) of melted K and high K loading content (97 wt. %); and (3) the 3-dimensional (3D) architecture with enough inner space effectively induces a homogeneous electric field and smooth K plating, enabling suppressed dendrite and impedance growth during cycling. Hence, these porous N-doped carbon nanofibres along with monodispersed Zn clusters are obtained, and the corresponding structure-activity relationship of the K metal host is thoroughly discussed.

## Results

### Synthesis and physicochemical characterization of MSCNFs

The preparation of MSCNFs is shown in Fig. 1a. First, a nanosized MOF crystal, nanoMET-6 ([Zn-$(C_2N_3H_2)_2$])[57], was synthesized via a coordination reaction between zinc chloride and 1$H$-1,2,3-triazole at $25 \pm 2$ °C. Surfactant F-127, a block copolymer (PEO100-PPO65-PEO100), was employed as a capping agent to control the growth of MET-6. Due to the coordination effect between the hydrophilic groups of F-127 and the $Zn^{2+}$ sites on the MOF surface, the growth rate was slowed, and MET-6 nanocrystals with a particle size of ~50 nm (Supplementary Figs. 1 and 2) were harvested[58]. Here, nanoMET-6 acts as a pore transformer and potassiophilic source for the fabrication of nanofibre hosts. After the incorporation of PAN, the fibrous MOF membrane was collected by electrospinning (Supplementary Figs. 3 and 4a). Thereafter, a controlled pyrolysis process was conducted under an argon atmosphere at 600 °C, during which the triazole linkers in the embedded MOF particles decomposed and produced gases, forming porous carbon nanofibres (Supplementary Fig. 4b).

Powder X-ray diffraction (PXRD) indicates that no crystalline phase was present in the MSCNFs (Fig. 1b). The porosity of the MSCNFs was explored by $N_2$ adsorption/desorption measurements. The typical type IV isotherm and corresponding pore size distribution in Fig. 1c indicate that there were mainly mesopores (>4 nm) and partial micropores

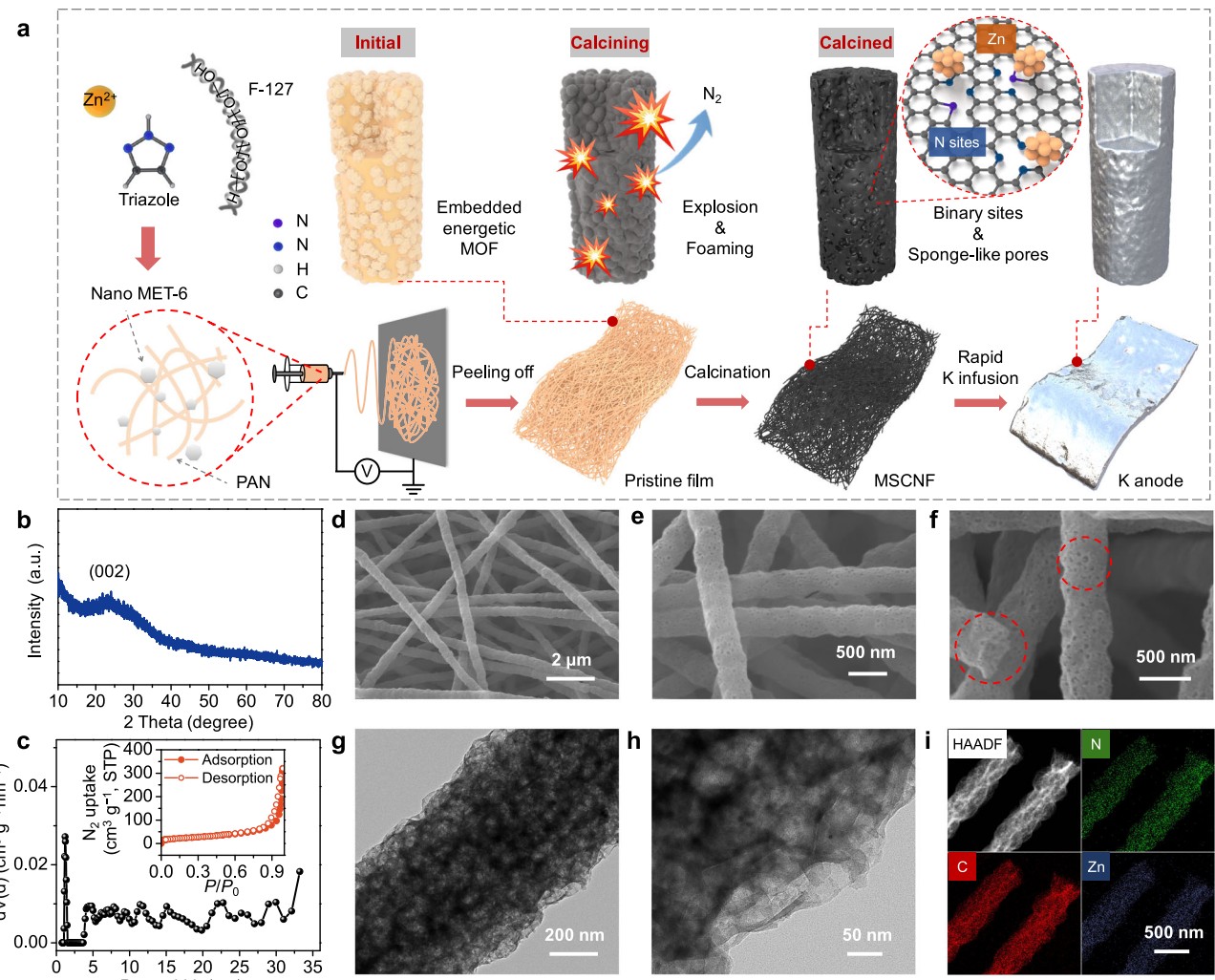

**Fig. 1 | Synthesis and physicochemical characterization of MSCNFs. a** Synthetic route of MSCNFs and the corresponding composite K anode. **b** PXRD patterns of MSCNFs. **c** Pore distribution of MSCNFs and the corresponding $N_2$–77 K adsorption isotherms (inset). **d**–**f** SEM images of MSCNFs. The red dashed lines highlight the porous structure of the MSCNFs. **g** TEM image and **h** HRTEM image of MSCNFs. **i** EDS elemental mapping of MSCNFs in HAADF mode.

(1.2 nm) in the MSCNFs, reflecting a hierarchically porous structure. The specific surface area of MSCNFs was calculated to be 82 $m^2\,g^{-1}$ based on the Brunauer–Emmett–Teller method (Supplementary Table 1). As Fig. 1d–f and Supplementary Fig. 5 show, the as-synthesized MSCNFs exhibited a uniform size of ~500 nm in diameter with a homogeneous distribution of mesopores (<50 nm) outside and inside the fibers. Transmission electron microscopy (TEM) images (Fig. 1g) confirm that the closely packed nanovoid space constituted an interconnected hollow structure in the MSCNFs. Optical images reveal that the MSCNFs still retained ~75% of their original size after pyrolysis (Supplementary Fig. 4), well inheriting the flexibility of MOF/PAN pristine films (Supplementary Fig. 6). In addition, benefiting from the electrospinning method and its intrinsic highly porous features, the MSCNFs exhibited a surface density of ~0.5 mg $cm^{-2}$ with a thickness of 100 μm (Supplementary Fig. 7).

High-resolution TEM images (Fig. 1h) and Raman spectroscopy measurements (Supplementary Fig. 8) further confirm the amorphous feature of MSCNFs. To verify the elemental composition of MSCNFs, elemental mapping under high-angle annular dark-field mode was utilized, which reveals the uniform distributions of C, N, and Zn (Fig. 1i). X-ray photoelectron spectroscopy (XPS) validates that the deconvoluted Zn 2p spectra show characteristic peaks of Zn $2p_{3/2}$ (1021.6 eV) and Zn $2p_{1/2}$ (1044.7 eV), corresponding to Zn metal (Supplementary Fig. 9a)[59]. The N 1s spectra (Supplementary Fig. 9b) depict

the existence of pyridinic N (398.7 eV), pyrrolic N (400.1 eV), and graphitic N (403.3 eV). The quantification analysis of XPS (Supplementary Table 2) shows that the contents of C, N, Zn in MSCNFs were 60.1%, 15.2%, and 17.8% (weight ratio), respectively. The above results imply that (1) the foaming effect well established hierarchical pores in MSCNFs; (2) a sufficient nitrogen source from PAN and MET-6 provided a high percentage of N-doped carbon; and (3) with Zn ions being well confined in the coordinated frameworks, the as-reduced Zn products preserved their high dispersity and small size, which becomes the key factor for the generation of Zn clusters.

## Investigations on the MSCNF potassiophilicity

The potassiophilicity of the MSCNFs was first evaluated by an electrochemical K plating test of asymmetric K||MSCNF half-cells (the typical behavior of K nucleation on a host is defined in Supplementary Fig. 10a). Under a current density of 0.5 mA $cm^{-2}$, the MSCNFs exhibit a rather flat voltage plateau upon discharge, corresponding to a low nucleation overpotential of <15 mV (Fig. 2a). In comparison, the K deposition on bare Cu foil shows a significant voltage dip (nucleation overpotential of 133 mV) at the beginning, which is negligible in the K|| MSCNF cell. This result demonstrates that MSCNFs are favorable for K deposition. Different from the nonporous Cu substrate, MSCNFs exhibit a delay in nucleation, which can be ascribed to SEI formation

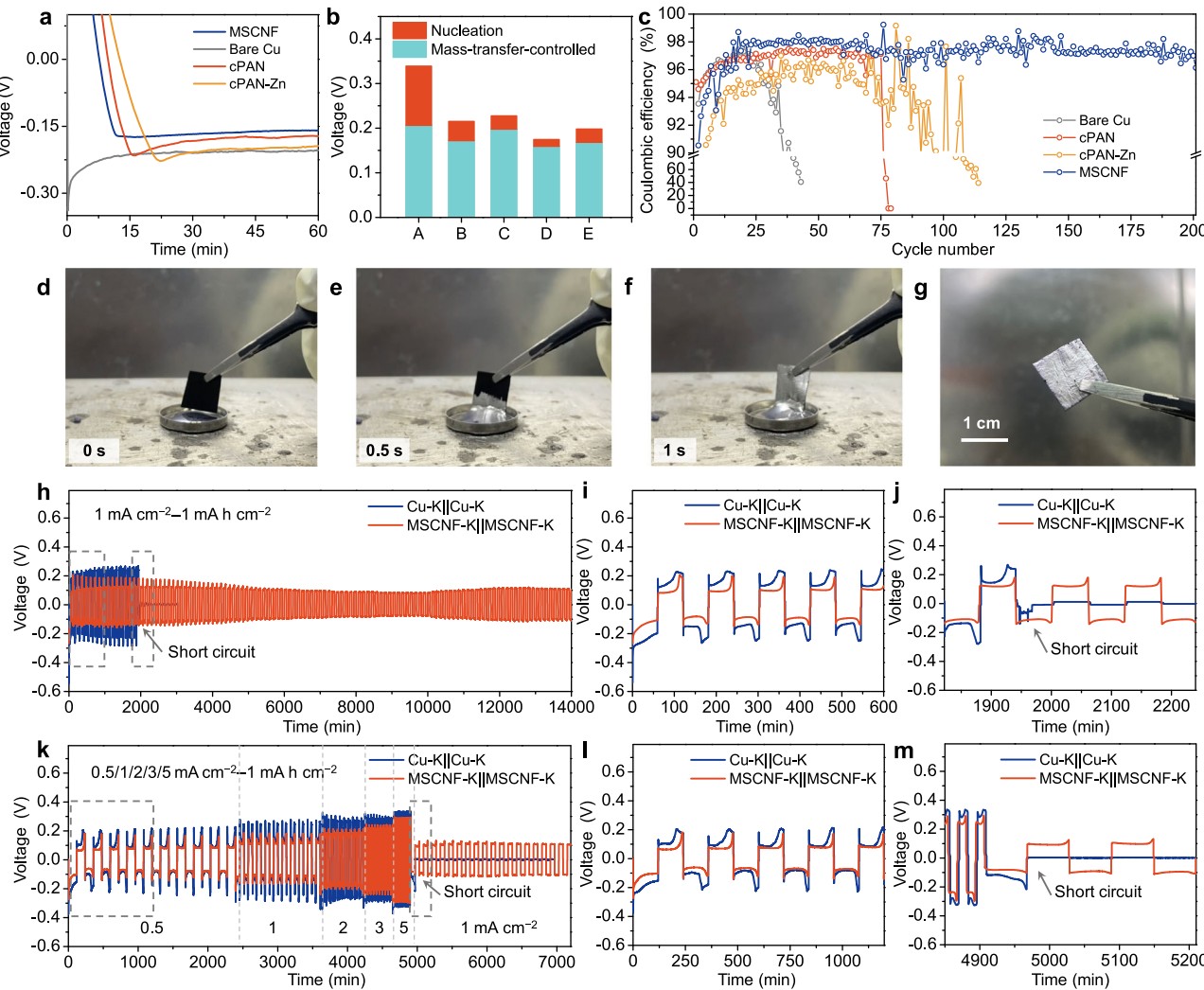

**Fig. 2 | Electrochemical characterizations of various potassium metal hosts.**
**a** Voltage profiles of K nucleation on different hosts at a current density of
0.5 mA cm$^{-2}$. **b** Overpotentials of K deposition on different hosts (A: bare Cu, B:
cPAN, C: cPAN-Zn, D: MSCNF, E: etched MSCNF). **c** Coulombic efficiencies of K
plating/stripping on different hosts at a current density of 0.5 mA cm$^{-2}$. **d**–**g** Optical
images of K infusion into MSCNFs and the corresponding product. **h**–**j** Voltage
profiles of symmetric cells with MSCNF-K and Cu-K anodes and corresponding
enlarged profiles of the marked regions. **k**–**m** Rate performances of different
symmetric cells at different current densities and corresponding enlarged profiles
of the marked region. All electrochemical measurements were carried out at a
temperature of 25 ± 2 °C.

during the first K uptake, as the delay disappears in subsequent cycles
(Supplementary Fig. 10b)[41,42,60,61]. To further confirm the contributions
of different components to this potassiophilicity, other control samples, including carbonized PAN fibers (cPAN), carbonized PAN fibers
decorated with Zn metal particles (cPAN-Zn) and etched MSCNFs (Zn
removed), were prepared. As the XPS spectra in Supplementary
Fig. 11a–d reveal, cPAN and cPAN-Zn contain a composition of N similar
to that of MSCNFs, and Zn is maintained in cPAN-Zn. Unlike the
MSCNFs, cPAN and cPAN-Zn are relatively nonporous on account of
their low surface areas (Supplementary Fig. 12) and morphologies
(Supplementary Figs. 13–15). The PXRD patterns illustrate the amorphous features of cPAN and cPAN-Zn (Supplementary Fig. 16). As
shown in Fig. 2a, the voltage curves of cPAN and cPAN-Zn present
voltage dips during discharge, and their nucleation overpotentials are
43 and 30 mV, respectively. Etched MSCNFs, interestingly, deliver a
slightly higher nucleation overpotential (30 mV) than that of MSCNFs
with an observable voltage dip (Supplementary Fig. 17). Based on the
deposition behaviors of different electrodes, their performance is
summarized in Fig. 2b, which indicates that (1) MSCNFs realize the
lowest nucleation and mass-transfer-controlled overpotential, proving
their good potassiophilicity and charge transfer, whereas the effect

weakens after Zn removal; (2) compared to the potassiophobic Cu foil,
cPAN has better potassiophilicity with the help of N-doped sites, but its
improvement is limited; and (3) for cPAN-Zn, the nucleation overpotential can be further reduced by Zn incorporation, while the
agglomerated Zn nanoparticles (Supplementary Fig. 14) bring harm to
the charge transfer in the carbon network. Next, the CE (the ratio of K
stripping to K plating) of the selected hosts was investigated (Fig. 2c).
All half cells (i.e., where K metal is used as a negative/counter electrode) were allowed to undergo five formation cycles before the galvanostatic cycling test (Supplementary Fig. 18) to stabilize the SEI
before plating and remove any electrochemically unstable residual
impurities[41,42]. Due to SEI formation, all the hosts experience relatively
low CEs in the initial several cycles[29]. The Cu foil exhibits a CE plunge
only after 40 cycles, which might be due to the rapid dendrite growth
and repeated breakdown and reconstruction of the SEI[38]. In comparison, the MSCNFs show a higher CE (>97%) after 10 cycles and maintain
good performance for over 200 cycles. However, it is noteworthy that
a CE of 97% is still far from satisfactory for practical applications. To
achieve a high CE for practical applications (>99.99%), the priority is to
avoid the irreversible loss of active K during cycling, which means
strategies should be focused on preventing the further decomposition

of electrolytes and the excessive or continuous formation of the SEI. Briefly, apart from a highly potassiophilic and robust host, a sufficient CE for the practical application of K metal batteries could be realized via a rational electrolyte design that combines the following features: (1) suitable anions, such as FSI⁻, for F-rich SEI formation; (2) solvent with a low dielectric constant and higher value of the lowest unoccupied molecular orbital energy; and (3) high-concentration electrolytes or localized high-concentration electrolytes (aggregated solvent and anions) that provide a stable K⁺-solvent structure[41,62].

K metal thermal infusion experiments were carried out to further assess the potassiophilicity from chemical and physical perspectives. As the photographic pictures (Fig. 2d–g) and movie (Supplementary Movie 1) illustrate, the MSCNF film with an area of $1 \times 1$ cm² can realize a homogeneous K infusion once the edge of the film makes contact with melted K (150 °C). The whole process takes less than 1 s to complete, which is one of the shortest times reported in the literature for K infusion (Supplementary Table 3). This peculiar potassiophilicity likely benefits from the synergistic effect of the potassiophilic sites and the capillary force guided by the hierarchical pores and gaps[45,63]. Moreover, the obtained MSCNF-K film can be reversibly bent and recovered under an external force, confirming the maintained flexibility of the MSCNFs (Supplementary Fig. 19). Cu foil, cPAN, and cPAN-Zn films with the same size were also examined by placing them in contact with melted K (Supplementary Movies 2–4). However, the three hosts, especially the Cu foil, exhibit poor philicity towards K (Supplementary Fig. 20). cPAN and cPAN-Zn show partial intake with inhomogeneous K distribution (Supplementary Figs. 21 and 22). Furthermore, an exfoliation phenomenon can be observed from the side view of the K-infused cPAN-Zn film (Supplementary Fig. 21d), indicating that K is unable to homogeneously infiltrate the nonporous substrate without strong capillary force. This again demonstrates the enhanced potassiophilicity of the MSCNFs. Then, the as-obtained MSCNF-K shows a density of ~15.5 mg cm⁻² (K loading >96 wt. %) and was tested as a negative electrode active material for nonaqueous K-ion storage. By stripping K metal in the MSCNF-K||Cu cell at a current density of 0.05 mA cm⁻², the total capacity of MSCNF-K is calculated to be 667 mA h g⁻¹, close to the theoretical capacity of pure K metal (Supplementary Fig. 23 and Supplementary Table 3). In addition, the stripped electrode shows structural integrity without any K metal impurities, demonstrating good reversibility (Supplementary Fig. 24). To unveil the long-term galvanostatic cycling capability of MSCNF-K, symmetric cells of MSCNF-K||MSCNF-K were tested under a current of 1 mA cm⁻² and cut-off capacity of 1 mA h cm⁻². As shown in Fig. 2h, the MSCNF-K||MSCNF-K-cell realizes a low average overpotential of ~100 mV and stable cycling for over 800 h. The gradual decrease in the initial cycles could be ascribed to surface activation (Supplementary Fig. 25)[45,64]. As a comparison, a symmetric Cu-K (prepared by mechanically pressing the K metal on the top of the Cu foil) cell shows larger polarization during cycling (>150 mV), and it steadily increases until a short circuit occurs after ~32 h (Fig. 2i and j). The rate performance of these symmetric cells was also examined. With the current density switched from 0.5 mA cm⁻² to 5 mA cm⁻², MSCNF-K||MSCNF-K guarantees stable plating and stripping with a lower overpotential than that of Cu-K||Cu-K (Fig. 2k and l). When the current returns to 1 mA cm⁻², the voltage curve of MSCNF-K||MSCNF-K changes smoothly, while Cu-K||Cu-K exhibits an immediate short circuit (Fig. 2m). Cycling tests under larger cut-off capacities of 3 and 5 mA h cm⁻² also demonstrate the reversibility of MSCNF-K electrodes, whereas Cu-K fails to function once the tests start, indicating possible dendrite issues (Supplementary Figs. 26 and 27).

## Investigation of potassium uptake in MSCNFs via operando, in situ, and ex situ physicochemical measurements

To reveal the origin of the electrochemical performance of MSCNFs, in situ characterizations, along with ex situ SEM, were conducted. In situ optical microscopy was first applied to visualize the surface morphology evolution of different hosts during K plating. As the amount of plated K increases, mossy-like dendrites appear on the surface of Cu foil only after 3 min, and they quickly become conspicuous in upstanding rod shapes. At the end of the plating experiment, the Cu surface is fully covered by rod-like K dendrites (Fig. 3a). In sharp contrast, the surface of the MSCNFs remains unchanged throughout the plating process, implying that MSCNFs can effectively induce homogeneous K deposition without agglomeration and therefore achieve a dendrite-free K anode (Fig. 3b). Operando XRD (Supplementary Fig. 28) discloses a similar behavior (Figs. 3c and 3d): when the plating process is launched, the diffraction peaks at 23.6° and 41.4°, corresponding to the (110) and (211) plane of K metal (PDF no. 01-0500), respectively, show a lag for the Al/MSCNF||K cell compared to those of Al||K, which means that there is a lower exposure of K metal in the Al/MSCNF electrode during the early plating stage (an 18-µm-thick Al foil was selected both as the current collector and the window at the cathode side for its proper mechanical strength and similar K deposition behavior to the Cu foil).

The morphologies of the plated hosts were further monitored using ex situ SEM measurements. Cu||K and MSCNF||K half cells at different discharge states were disassembled, and the host samples were collected after proper treatment (for details, see Methods). For the Cu foil (Supplementary Fig. 29), its surface forms loosely stacked microsized particles once it is discharged to 0.5 mA h cm⁻². As the discharge proceeds, mossy fiber-like dendrites appear and become larger, leading to a short circuit at a capacity of 5 mA h cm⁻² (Supplementary Figs. 29 and 30). For the MSCNFs, however, the phenomenon differs (Fig. 3e and f): a discharge at 0.5 mA h cm⁻² causes the homogeneous filling and coverage of K on the porous fibers. Thereafter, the plated fibers thicken with extra K metal, start to cover the gaps between fibers and then establish a flat and smooth surface (Supplementary Fig. 31). The cross-section view of the deposited Cu electrode confirms the uneven surface and the loose structure of K after deposition (Supplementary Fig. 32). The thicknesses of K deposited on the Cu foil are ~15 and ~84 µm at discharge capacities of 0.5 and 3 mA h cm⁻², respectively, which are much larger than the corresponding calculated values of 8.5 and 50.7 µm (for details, see Supplementary Fig. 32 panels and caption). In contrast, the plated MSCNFs present a flat appearance, showing thicknesses of ~59 and ~58 µm at 0.5 and 3 mA h cm⁻², respectively (Supplementary Fig. 33). At 0.5 mA h cm⁻², some aggregated K particles form and uniformly distribute within the entirety of the MSCNF host (Supplementary Fig. 33a). In contrast, there are no recognizable aggregates on the surface of the MSCNF host. Additionally, cross-sectional EDS elemental mappings of the plated MSCNFs in different K uptake states confirm the homogeneous distribution of K in the deposited host (Supplementary Figs. 34 and 35). Moreover, the MSCNF electrode discharged at 5 mA h cm⁻² gives a thickness of ~90 µm (Supplementary Fig. 36a), which is close to the calculated value of 84.4 µm (see Supplementary Fig. 32 caption). Its cross-section image shows a relatively smooth surface and uniform distribution of carbon nanofibres (Supplementary Fig. 36b). The above phenomena signify that (1) due to a lack of effective guidance and space confinement, the K metal deposited on Cu foils tends to form a loose and porous film and (2) the MSCNFs provide distinct regulation for homogeneous K deposition that guarantees a dense and flat metal composite film.

## Computational investigations into the K deposition mechanisms in various hosting structures

Density functional theory (DFT) was applied to verify the interactions between K atoms and different hosts. K atoms bonded to carbon (K-C), N-doped carbon (K-NC) and N-doped carbon decorated with Zn clusters (K-NC-Zn) were modeled for comparison (Fig. 4a–c). As shown in Fig. 4d, K-C displays the lowest binding energy ($E_b$) of 0.62 eV among

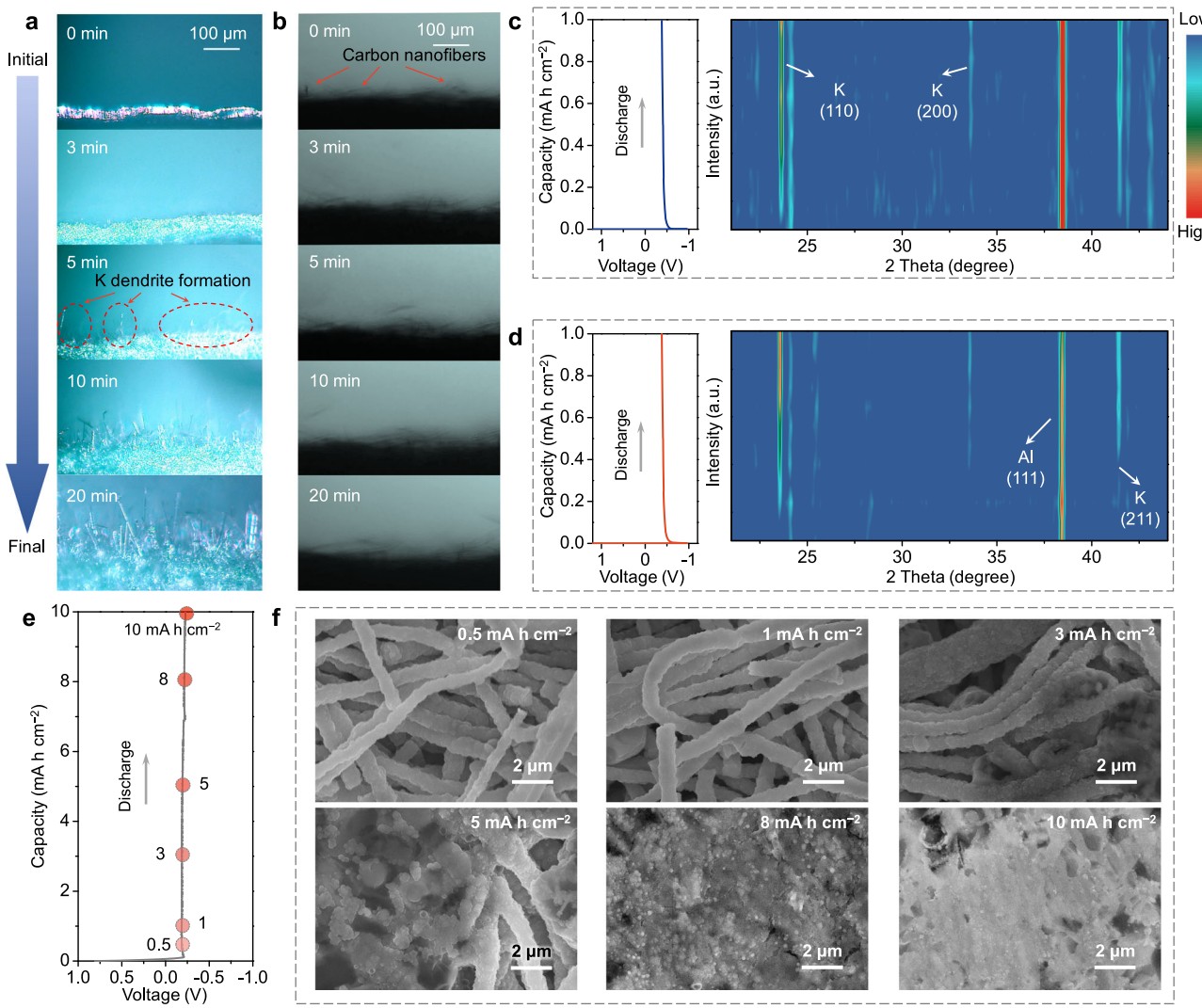

**Fig. 3 | Physicochemical investigation of potassium metal deposition for various potassium metal hosts.** In situ optical microscopy observation of K deposition on **a** Cu foil and **b** MSCNFs at a current density of 6 mA cm⁻². Contour plots of operando XRD patterns for K deposition on **c** Al foil and **d** MSCNFs at a current density of 0.5 mA cm⁻². **e** Voltage profiles of K deposition on MSCNFs at a current density of 0.5 mA cm⁻² and **f** the corresponding SEM images of deposited MSCNFs at different discharge states. All electrochemical measurements were carried out at a temperature of 25 ± 2 °C.

the three models. With the incorporation of N sites, the $E_b$ of K-NC is improved (2.05 eV), consistent with previously reported results[48]. K-NC-Zn displays the highest binding energy of 2.94 eV, demonstrating that the cooperation between N sites and Zn clusters can result in the strongest interaction with K atoms. This advantage might be related to the existence of the alloy compound KZn₁₃ at room temperature[65,66]. In addition, model cells were constructed to simulate the electric field in nonporous carbon fiber (CF) and MSCNF electrodes (Fig. 4e–h). The cross-section view of the E-field distribution of the CFs can be seen in Fig. 4f. The top layer of the CFs towards the cathode shows an obvious E-field polarization, which causes a shielding effect. In contrast, the MSCNF host delivers a much more homogeneous E-field distribution than the CF host (Fig. 4h). This phenomenon is due to the reduced electrical conductivity of the defective host and the porous structure that increases the electron transport barrier within the electrode, further illustrating the advantage of MSCNFs[67,68].

Therefore, the computational results together with the morphology characterization demonstrate that MSCNFs are good at realizing uniform K nucleation and deposition by providing homogeneous potassiophilic sites and regulating the K⁺ flux within the host structure. Specifically, we speculate that the K plating process can be divided into three phases (Figs. 4i and 4j; Supplementary Fig. 37):

Phase I (Fig. 4i), wherein K metal starts to nucleate inside the pores of MSCNFs, finally filling up the pores and fully covering the carbon nanofibres without causing a volume change of MSCNFs (<0.5 mA h cm⁻²); Phase II, wherein the metal cover on the nanofibres continues to grow and fills up the voids between the carbon nanofibres, forming a dense composite film (<3 mA h cm⁻²); and Phase III, wherein an extra amount of K metal continues to deposit on the MSCNF-K and thicken the film, leading to a volume change that follows the theoretical value. In comparison, a bare Cu foil tends to cause inhomogeneous K nucleation and the uncontrollable formation of porous K film and K dendrites (Fig. 4j).

## Electrochemical and physicochemical characterizations of the K metal and MSCNF-K electrodes

Electrodes harvested from symmetric cells at different cycling periods (pristine, 1 cycle, 10 cycles and 50 cycles) were also investigated by ex situ SEM measurements. Figure 5a–d and Supplementary Fig. 38 illustrate the morphology evolution of the bare K surface as cycling proceeds, which changes from a smooth yet dense state to a loose construction with cracks and voids. The surface of the MSCNF-K electrode prepared via the thermal infusion method retains traces of immersed fibers (Fig. 5e; Supplementary Fig. 39a); however, the

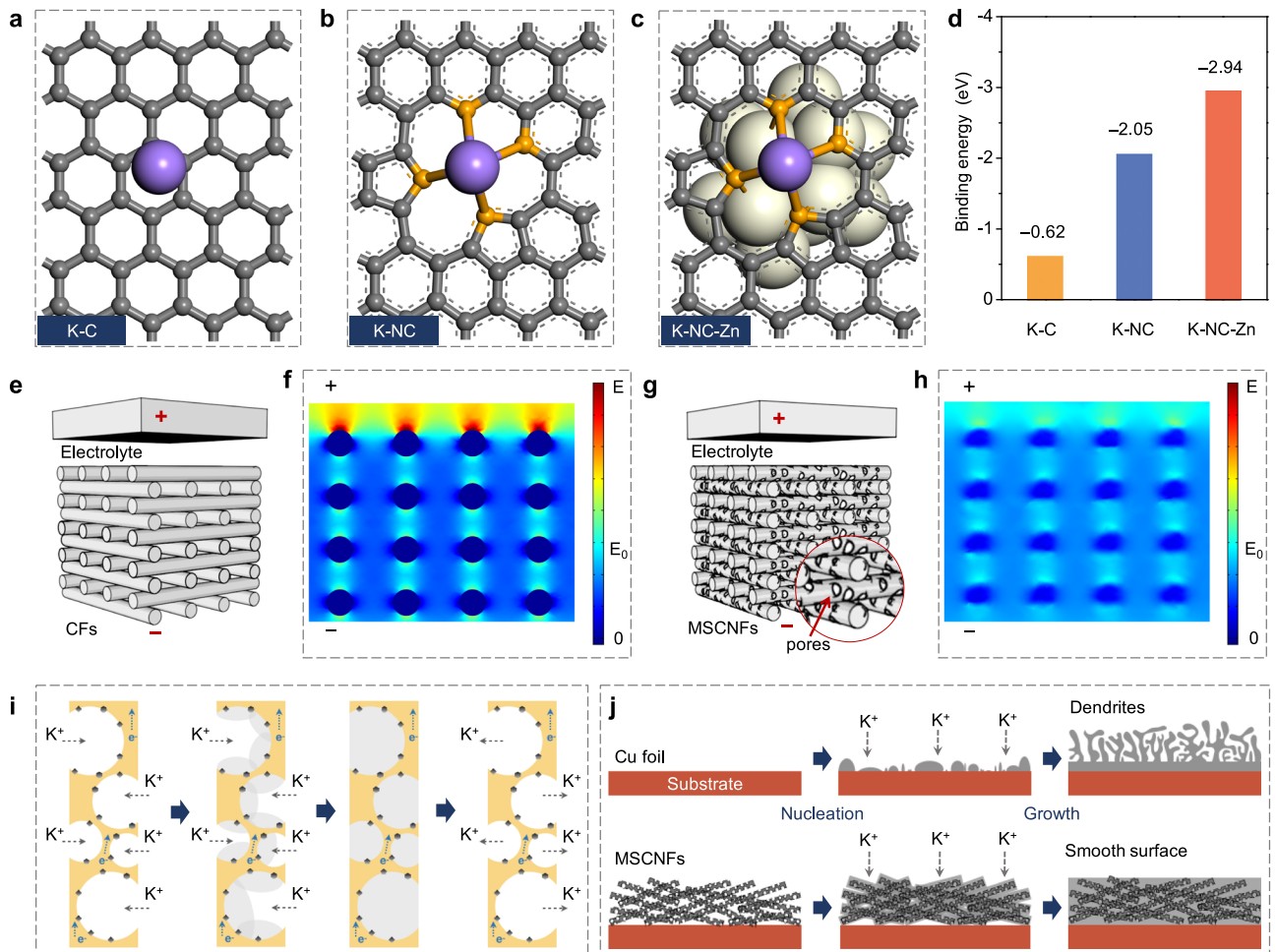

**Fig. 4 | Theoretical investigation of potassium metal deposition for various potassium metal hosts. a** Calculated binding energies of K atoms with different hosts by DFT and **b**–**d** corresponding models. Simulation models of **e** CFs and **g** MSCNFs and corresponding E-field distributions of the **f** CFs and **h** MSCNFs. Schematic illustration of **i** K plating and stripping on MSCNFs and **j** K deposition on different substrates.

morphology is relatively flat overall. After cycling, MSCNF-K maintains its integrity well, and no defects emerge on the electrode surface (Fig. 5f–h; Supplementary Fig. 39b–d). Chemical information on the cycled electrode (after 1 cycle and 20 cycles) was further obtained by XPS analysis (Fig. 5i–n). C 1$s$, O 1$s$ and F 1$s$ spectra of the Cu-K and MSCNF-K electrodes were collected, and their peaks were fitted. The fresh K and MSCNF-K electrodes show identical compositions, and the existence of C = O (-289 eV) and C-O (-286 eV) is attributed to the formation of $K_2CO_3$ on the electrode surface during sample preparation[45]. Upon cycling, the chemical composition varies between the two kinds of electrodes: (1) for Cu-K, the relative intensities of C-C (284.5 eV) and S = O (532.8 eV) bonds related to the decomposition of the electrolyte drastically change, along with the absence of S-F (683.9 eV) in the 1st cycle; and (2) for MSCNF-K, most of the characteristic peaks remain unchanged throughout cycling. Moreover, the peaks shift slightly as the cycle number increases, which can be due to the thickening of the SEI and changes in the electrical conductivity of the samples[42]. Consequently, we can conclude that the Cu-K anode experiences serious surface damage during cycling due to a volume change, which leads to an unstable SEI and irregular K deposition. MSCNF-K is able to realize an intact smooth surface because of the suppressed volume change; therefore, a stable and homogeneous SEI is guaranteed. In return, this SEI regulates and facilitates a uniform ion flux, further enhancing the stability of the composite anode. Tafel curves of the symmetric cells were also gathered via linear sweep voltammetry (LSV). Notably, the MSCNF-K electrode produces a larger exchange current ($4.2 \times 10^{-6}$ A cm$^{-2}$) than the Cu-K electrode ($1.9 \times 10^{-6}$ A cm$^{-2}$), confirming the better capability of MSCNFs to suppress dendrite growth and stabilize the SEI film (Supplementary Fig. 40)[69].

## Electrochemical energy storage performances of K metal and MSCNF-K electrodes in various full cell configurations

To evaluate the performance of MSCNF-K as an anode material, full cells comprising $Se_{0.05}S_{0.95}$@pPAN cathodes (Supplementary Fig. 41) and MSCNF-K anodes were assembled and tested. $Se_{0.05}S_{0.95}$@pPAN (the chemical composition was determined by elemental analysis, see Supplementary Table 4) was considered a positive electrode active material because of its demonstrated ability to store alkali metal ions in LIBs and Na-ion batteries (NIBs)[70,71]. Cu-K||$Se_{0.05}S_{0.95}$@pPAN cells were also assembled and tested for comparison. The rate performance of K–S batteries was first assessed. Figure 6a shows that the two kinds of K–S batteries exhibit identical initial discharge capacities (-960 mA h g$^{-1}$ at the 2nd cycle) based on the weight of selenium-doped sulfur. However, the MSCNF-K||$Se_{0.05}S_{0.95}$@pPAN cell displays a lower voltage hysteresis between charge and discharge (0.41 V) than Cu-K||$Se_{0.05}S_{0.95}$@pPAN (0.52 V) at 0.1 A g$^{-1}$, and it retains higher and more stable capacity outputs of 891, 738, 616, 469 and 363 mA h g$^{-1}$ with average discharge voltages of 1.59, 1.53, 1.47, 1.36 and 1.32 V (calculated from the ratio of discharge energy to discharge capacity of a cell) at current rates of 0.2, 0.5, 1, 2, and 3 A g$^{-1}$, respectively (Fig. 6b, c). In contrast, the Cu-K||$Se_{0.05}S_{0.95}$@pPAN cell exhibits

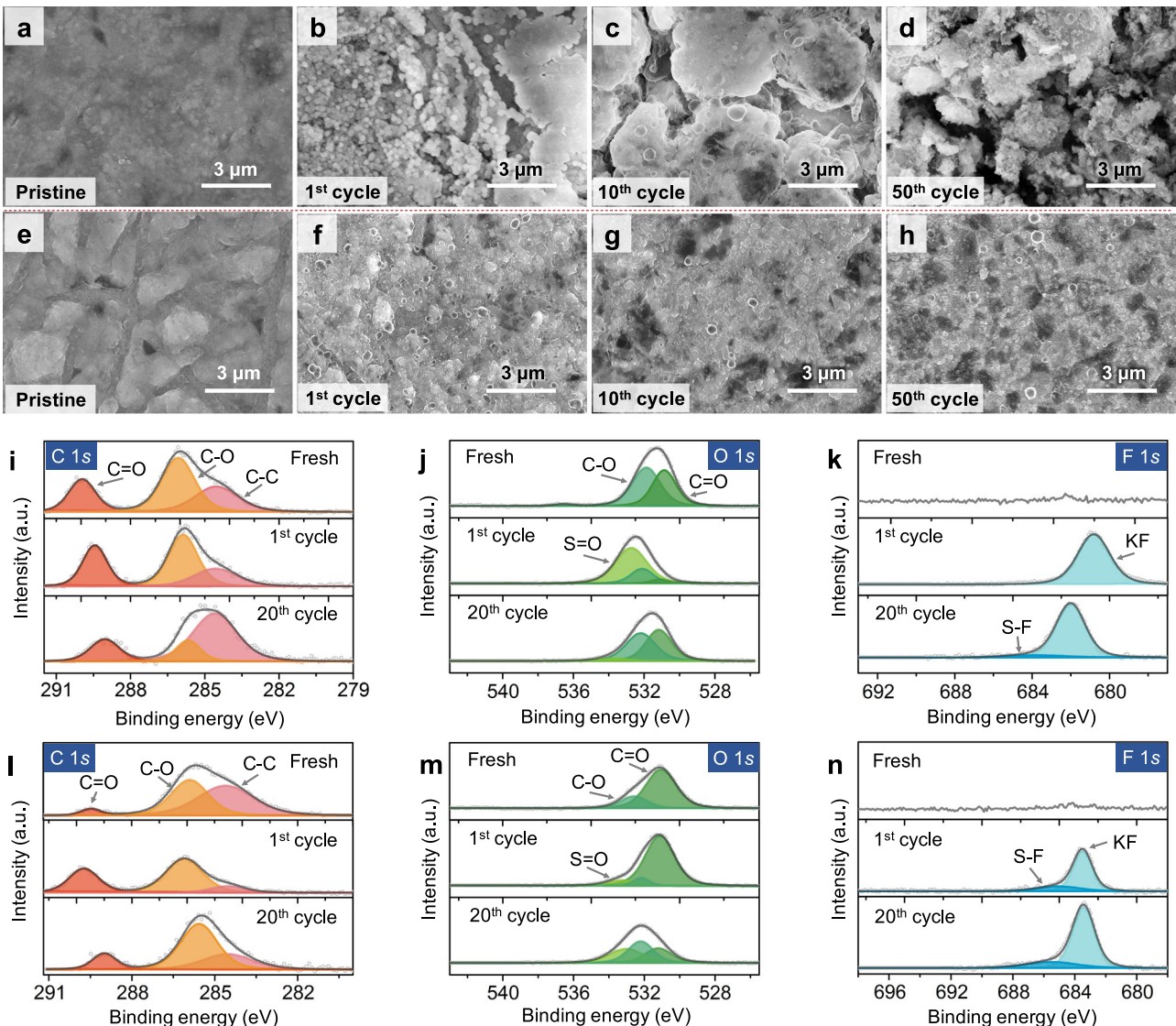

**Fig. 5 | Ex situ physicochemical characterizations of various potassium metal-based electrodes.** Ex situ SEM images of **a**–**d** bare K and **e**–**h** MSCNF-K anodes from symmetric cells at different cycle numbers. Ex situ XPS spectra of **i** C 1*s*, **j** O 1*s* and **k** F 1*s* from symmetric cells based on Cu-K anodes at different cycle numbers (Fresh, after the 1st charge, and after the 20th charge). Ex situ XPS spectra of **l** C 1*s*,

**m** O 1*s* and **n** F 1*s* from symmetric cells based on MSCNF-K anodes at different cycle numbers (Fresh, after the 1st charge, and after the 20th charge). The ex situ electrodes were sampled at an intermediate state of potassiation. All electrochemical measurements were carried out at a temperature of 25 ± 2 °C.

discharge capacities of 712, 626, 498, 280, 168 mA h g⁻¹ with average discharge voltages of 1.59, 1.52, 1.39, 1.31 and 1.30 V at current rates of 0.2, 0.5, 1, 2, and 3 A g⁻¹, respectively. The capacity of MSCNF-K||Se$_{0.05}$S$_{0.95}$@pPAN recovers once the specific current is switched back to 0.2 A g⁻¹, whereas Cu-K||Se$_{0.05}$S$_{0.95}$@pPAN suffers from a steady capacity drop (see Fig. 6a). Long-term cycling tests at 500 mA g⁻¹ (Fig. 6c) also disclose that the discharge capacity of Cu-K||Se$_{0.05}$S$_{0.95}$@pPAN drops rapidly, showing a capacity retention of 60% after only 120 cycles (calculated from the fourth cycle to the final cycle under the specific current of 500 mA g⁻¹; three precycles at 100 mA g⁻¹ were conducted before cycling). Moreover, the cell with the MSCNF-K anode delivers a significantly enhanced stability, which can retain 60% of the initial capacity for over 600 cycles (Supplementary Fig. 42). Such a difference can be attributed to the continuous dendrite growth and electrolyte consumption during the Cu-K anode's plating/stripping process, while the MSCNF-K anode is more resilient. To further investigate the compatibility of MSCNF-K with other types of cathodes, a typical intercalation-type cathode, K$_{0.220}$Fe[Fe(CN)$_6$]$_{0.805}$ (Potassium Prussian blue, PPB, Supplementary Fig. 43), was prepared and paired

with our K anodes to obtain K–PPB full cells. MSCNF-K||PPB and Cu-K||PPB cells show identical galvanostatic charge/discharge profiles with a voltage plateau between 3.1–3.4 V (Supplementary Fig. 44) at a specific current of 50 mA g⁻¹. However, the MSCNF-K||PPB cell delivers a lower voltage hysteresis between charge and discharge at 50 mA g⁻¹ (0.26 V) than Cu-K||PPB (0.39 V) and retains a more specific discharge capacity at higher current rates (Supplementary Figs. 44 and 45), which is similar to the trend in K–S batteries. This result confirms the favorable electrochemical performance and compatibility of MSCNF-K anodes in different electrochemical energy storage systems.

In situ electrochemical impedance spectroscopy (EIS) was carried out to investigate the interfacial evolution of different K–S batteries. The EIS spectra were collected during the cell discharge/charge process (the 2nd cycle) with short intervals, and the raw data were fitted using an equivalent circuit (Supplementary Fig. 46) and summarized accordingly (Supplementary Table 5)[72]. As shown in Fig. 6d–f, the Cu-K||Se$_{0.05}$S$_{0.95}$@pPAN cell experiences a notable increase in the SEI impedance ($R_f$) from 51.6 Ω at the initial state to 109.1 Ω at the beginning of recharge. Moreover, after recharge, the final $R_f$ becomes

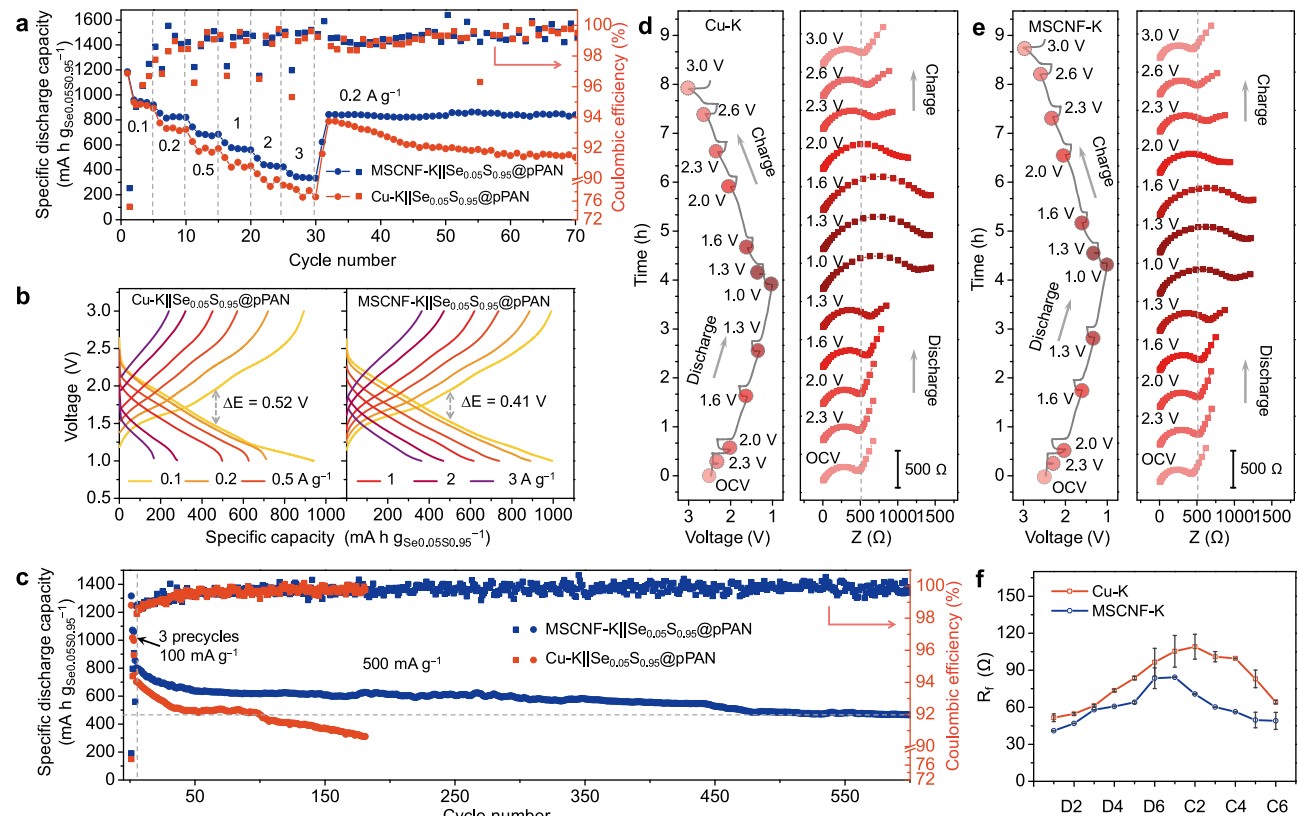

**Fig. 6 | Electrochemical energy storage performance of potassium-sulfur cells.** **a** Rate performance of full cells comprising Cu-K and MSCN-K negative electrodes. Voltage profiles of cells comprising **b** Cu-K and MSCNF-K negative electrodes at different current rates. **c** Long-term cycling performance at a specific current of 500 mA $g_{Se0.05S0.95}^{-1}$. In situ EIS results of **d** Cu-K ∥ $Se_{0.05}S_{0.95}$@pPAN and **e** MSCNF-

K∥$Se_{0.05}S_{0.95}$@pPAN. **f** $R_f$ values from the in situ EIS results at different states (D2: discharged at 2.4 V; D4: discharged at 1.6 V; D6: discharged at 1.0 V; C2: recharged at 1.6 V; C4: recharged at 2.3 V; C6: recharged at 3.0 V), where the error bars represent the standard deviation.

64.2 Ω, implying irreversible dendrite growth and SEI deterioration on the K anode. However, the K−S cells employing the MSCNFs-K anode exhibit an $R_f$ increase from 41 Ω to 49 Ω within a single cycle, and the value of $R_f$ at the initially recharged state (70.7 Ω) is also reduced compared to that of Cu-K, indicating good reversibility. Moreover, a similar trend takes place in the evolution of the charge transfer impedance ($R_{ct}$) during the whole cycle (Supplementary Fig. 47). Consistent with the symmetric cell results, EIS confirmed the favorable behavior of the MSCNF-K anode in terms of SEI stabilization and the regulation of K plating/stripping processes[48].

## Discussion

In summary, we have developed and characterized hierarchically porous carbon nanofibres (MSCNFs) with monodispersed binary active sites and N and Zn clusters. A series of electrochemical tests, computational analyses and in situ/ex situ characterizations revealed that this porous host together with Zn-containing binary potassiophilic sites provides the following merits: (1) high potassiophilicity that leads to the preferable nucleation of K atoms; (2) light weight and abundant space for high K metal accommodation; and (3) effective induction of a homogeneous electric field and smooth K plating. As a result, MSCNF-K composite anodes can be prepared by a fast thermal infusion (1 s for 1 cm²) process, demonstrating a high K loading content (97 wt. %) and specific discharge capacity (667 mAh $g^{-1}$ at 0.05 mA cm⁻² for the first cycle using a Cu foil used as the positive electrode). The as-obtained MSCNF-K electrodes also maintain a good rate capability and cycling stability in symmetric cell and K−S and K−PPB full cell tests. However, it should be noted that the porous features of the MSCNFs can hinder electrochemical energy storage

performance in the case of "anode-free" cells (e.g., cells where the negative electrode is formed in situ on the MSCNFs by electro-deposition on potassium sourced from the positive electrode) or K metal batteries with a low N/P ratio (e.g., <2). Indeed, in these systems (although not experimentally tested in the present manuscript), an MSCNF host would require large amounts of electrolyte, thus possibly decreasing the energy density of the cell. Accordingly, potential strategies that may reduce the requirements of electrolytes in an MSCNF-based battery would be (1) preparing thinner MSCNFs (<50 μm) via a proper electrospinning method to reduce the requirement of electrolyte[73], (2) pretreating the MSCNFs (controlled K deposition), partially lowering their surface area and suppressing interfacial side reactions with the electrolyte[74]. While the material design presented here represents a valid proof-of-concept for the production and application of carbon-based alkali metal hosting matrices, these findings also demonstrate the effectiveness of pore structure engineering in developing practical K metal batteries.

## Methods
### Materials
ZnCl₂ (99%) was purchased from 3Achem. F-127 (powder, average molecular weight $M_w$ = 126000 g mol⁻¹) and polyacrylonitrile (PAN, average molecular weight $M_w$ = 150000 g mol⁻¹), sulfur powder (99.99%) and selenium powder (99.99%) were purchased from Sigma Aldrich. Zn(CH₃COO)₂ (98%), N,N-dimethylformamide (DMF, 99.5%), HCl (36-38%), ammonium hydroxide (25-28%) and ethanol (99.7%) were purchased from SINOPHARM. Sodium carboxyl methyl cellulose (NaCMC), Super-P (99.9%), styrene butadiene rubber (SBR, 48%) and Cu foil (9 μm, 99.8%) were purchased from Kelude Experimental

Equipment Technology Corp. $FeCl_3 \cdot 6H_2O$ (99%) and polytetrafluoroethylene (PTFE, 60 wt. % dispersion in $H_2O$) were purchased from Aladdin. 1$H$-1,2,3-triazole (98%) was purchased from Energy Chemical. Carbon nanotubes (CNTs, diameter = ~10–20 nm, length = ~30–100 μm, 95%) were purchased from J&K Scientific. $K_4Fe(CN)_6$ (99%) was purchased from Macklin Inc. Aluminum foil (18 μm, 99.35%) was purchased from Kejing Star Technology Corp. All materials were used without further purification.

## Preparation of nanosized $Zn(C_2H_2N_3)_2$ (MET-6)

Typically, 10 mmol (2.0 g) of $ZnCl_2$ was first dissolved in a solvent mixture composed of 20 mL of DMF, 20 mL of ethanol, 30 mL of deionized water and 10 mL of ammonium hydroxide, forming solution A. A total of 43.2 mmol (2.5 mL) of 1$H$-1,2,3-triazole was mixed with 1.12 g F-127 powder to form solution B. Then, solution A was added dropwise to solution B, after which a white product was formed instantly. The suspension was stirred at slow speed for 2 h at 25 °C. The obtained white precipitate was then collected by centrifugation after washing (repeated dispersion and sonication in solvents) with DMF and ethanol 3 times. The wet powder was dried under vacuum for 24 h at 100 °C.

## Preparation of MSCNFs

First, 0.65 g nano-MET-6 was dispersed in 5 mL DMF solvent with sonication for 30 min. Then, 0.5 g PAN powder was added to the dispersion and stirred at 25 °C for 12 h. Then, the solution was transferred into a syringe and electrospun using an electrospinning machine at a high voltage of 15 kV. The nanofibres were collected with a piece of ground Al foil applied with a negative voltage of −5 kV. The distance between the needle and the Al foil was 15 cm. The flow rate of the precursor dispersion was maintained at 12 μL min⁻¹ controlled by a syringe pump. The experiment was conducted at 25 °C with 30% RH. The electrospun film was peeled off from the Al foil and transferred into an oven at 80 °C overnight. The film was calcined in an Ar-filled tube furnace first at 500 °C for 2 h with a ramping rate of 5 °C min⁻¹ and then at 600 °C for 1 h with a ramping rate of 5 °C min⁻¹.

## Preparation of etched MSCNFs

To obtain Zn-free porous carbon nanofibres, HCl solution was applied to remove the Zn species in the MSCNFs via the following reaction: $2HCl + Zn = ZnCl_2 + H_2$. Specifically, five circular pieces of MSCNFs with an area of 0.785 cm² were immersed in 20 mL of diluted HCl solution (3 M) for 1 h. Thereafter, the films were mildly rinsed with deionized water until the pH of the solvent became 7 and then dried under vacuum for 12 h at 100 °C.

## Preparation of cPAN

First, 0.5 g PAN powder was dissolved in 5 mL DMF with stirring at 25 °C for 12 h. Then, the solution was transferred into a syringe and electrospun using an electrospinning machine at a high voltage of 15 kV. The nanofibres were collected with a piece of grounded aluminum foil with an applied negative voltage of −5 kV. The distance between the needle and the aluminum foil was 15 cm. The flow rate of the precursor dispersion was maintained at 9 μL min⁻¹ by a syringe pump. The experiment was conducted at 25 °C with 30% RH. The electrospun film was peeled off from the Al foil and transferred into an oven at 80 °C overnight to dry. The film was first preoxidized at 260 °C in air for 1 h in a tube furnace with a ramping rate of 1 °C min⁻¹. After that, the film was calcined in an Ar atmosphere at 600 °C for 2 h with a ramping rate of 5 °C min⁻¹.

## Preparation of cPAN-Zn

$Zn(CH_3COO)_2$ (0.6 g) was first dissolved in 5 mL DMF solvent with sonication for 30 min. Then, 0.5 g PAN powder was added to the solution and stirred at 25 °C for 12 h. After stirring, the mixture was electrospun via the same method as cPAN. The electrospun film was peeled off from the Al foil and transferred into an oven at 80 °C overnight. The film was calcined in a vacuum tube furnace first at 500 °C for 2 h with a ramping rate of 5 °C min⁻¹ and then at 600 °C for 1 h with a ramping rate of 5 °C min⁻¹.

## K infusion experiments

The thermal infusion of K into the MSCNFs, cPAN and cPAN-Zn was conducted inside an Ar-filled glove box with water and oxygen contents both lower than 0.1 ppm. A piece of solid K metal (99%, Adamas-beta) was cleaned from impurities and melted in an inert stainless-steel case placed on a hot plate set at 150 °C. Subsequently, an edge of the $1 \times 1$ cm² sample was placed into contact with the molten K metal to start the infusion. After that, the composite was obtained after cooling to the ambient temperature of $25 \pm 2$ °C.

## Preparation of $Se_{0.05}S_{0.95}@pPAN$ and the corresponding cathode

Selenium and sulfur powder were mixed by ball milling in an Ar atmosphere at a molar ratio of 1:19 for 8 h with alcohol as a dispersant. A 50 mL PTFE-based cylindrical jar and zirconium-based spheres (diameter of 4 mm) were used for ball milling. The resulting mixtures were dried to remove the solvent and heated at 260 °C in an air oven for 12 h. The resulting $Se_{0.05}S_{0.95}$ was mixed with PAN at a weight ratio of 1:3 and then calcined at 300 °C for 2.5 h in an Ar atmosphere to collect the $Se_{0.05}S_{0.95}@pPAN$ composite. The electrode was prepared by mixing $Se_{0.05}S_{0.95}@pPAN$, Super-P, NaCMC and SBR at a weight ratio of 80:10:5:5 in deionized water using an agate mortar to form a slurry. Then, the slurry was coated onto an Al current collector and dried under vacuum at 50 °C for 12 h to form the electrode. The total mass loading and the active material mass loading of the electrode were ~2 and 0.8 mg cm⁻², respectively. The average thickness of the electrode was ~50 μm. The specific capacity of a K−S full cell was calculated based on the mass of Se and S in the cathode.

## Preparation of $K_{0.220}Fe[Fe(CN)_6]_{0.805}$ and the corresponding cathode

First, 20 mL of $FeCl_3 \cdot 6H_2O$ (2 mmol) aqueous solution was added to 80 mL of $K_4Fe(CN)_6$ (1 mmol) aqueous solution under stirring. Dark blue precipitate formed immediately and was collected after ageing for 24 h to obtain PPB powder. The powder was thoroughly washed with deionized water and ethanol by centrifugation and dried under vacuum at 80 °C for 12 h. PPB electrodes were fabricated by first rolling the ground $K_{0.220}Fe[Fe(CN)_6]_{0.805}$ powder, CNTs and PTFE in an agate mortar at a weight ratio of 7:2:1 to form a homogeneous mixture, which was then repeatedly compressed by a rolling machine 3-5 times to obtain a film with an average thickness of 100 μm. The total mass loading and the active material mass loading of the electrode were ~2 and 1.4 mg cm⁻², respectively. The specific capacity of a K-PPB full cell was calculated based on the mass of $K_{0.220}Fe[Fe(CN)_6]_{0.805}$ in the cathode.

## Material characterization

Powder X-ray diffraction (PXRD) was performed using a MiniFlex 600 diffractometer with a Cu-Kα X-ray radiation source (λ = 0.154056 nm). Operando XRD tests were carried out using the operando XRD battery cases obtained from Beijing Scistar Technology Co. Ltd. Field-emission scanning electron microscopy (FE-SEM) was performed using a Hitachi S-4800 operating at an accelerating voltage of 10.0 kV. Field-emission transmission electron microscopy (FE-TEM) and high-resolution TEM (HRTEM) were performed using an FEI Talos F200X. Nitrogen sorption isotherms were measured at 77 K using a Kubo-X1000 after pretreatment by heating the samples under dynamic vacuum at 150 °C for 12 h. X-ray photoelectron spectroscopy (XPS) was performed using a Thermo Scientific ESCALab 250Xi with 200 W monochromated Al Kα radiation. The discharged or recharged electrodes were thoroughly washed with dimethyl ether (DME) to remove the residual electrolyte

and salt before ex situ XPS and SEM measurements. Sealed containers specifically designed for SEM and XPS measurements were employed for sample transfer from the glove box to the instruments. The Raman spectrum was obtained using a Horiba LabRAM HE Evolution Raman spectrometer with a 532-nm laser.

## Cell assembly and electrochemical tests

CR2032 coin-type cells were assembled in a glove box filled with high-purity argon (water and oxygen contents below 0.1 ppm) and tested using a LAND-CT2001A tester. A solution of 1 M potassium bis(fluorosulfonyl)imide (KFSI) in ethylene carbonate/diethyl carbonate (EC/DEC, 1:1, v/v) with a water content of 4.4 ppm was utilized as the electrolyte (purchased from Dodo Chem). For K deposition tests, different hosts, including Cu foil, cPAN, cPAN-Zn, MSCNFs and etched MSCNFs, were assembled into coin cells with K metal foil as the counter/reference electrode and a glass fiber membrane (Whatman, GF/D, porosity of 2.7 μm, thickness of 675 μm) as the separator. All the asymmetric cells reported in this manuscript were activated by cycling within 1–0.01 V for 5 cycles at a current density of 0.1 mA cm$^{-2}$. The K plating/stripping performance was tested by plating K metal anodes onto different hosts at a current density of 0.5 mA cm$^{-2}$ with a cut-off capacity of 1 mA h cm$^{-2}$ and then stripped to 1 V. The specific capacity of the infused MSCNF-K electrode was tested by discharging the MSCNF-K||Cu coin cell at 0.02 mA cm$^{-2}$, whose value was determined based on the total mass of K metal and MSCNFs in the composite anode. The symmetric cells were assembled with two identical electrodes (Cu-K with Cu-K or MSCNF-K with MSCNF-K). The cells were galvanostatically tested at a current density of 1 mA cm$^{-2}$ and cut-off capacities of 1, 3 and 5 mA h cm$^{-2}$. The rate performance of the cells was assessed by cycling the cells at different current densities with a cut-off capacity of 1 mA h cm$^{-2}$. In situ optical observation was conducted on a homemade transparent cell by sealing a quartz cuvette (12.5 × 12.5 × 45 mm) with a cylindrical plug made of silicone rubber. The cell consisted of a K metal foil mechanically pressed on a Cu foil as the counter/reference electrode and a host (bare Cu foil or MSCNFs attached to Cu foil using conductive tape) as the working electrode, together with 3 mL of electrolyte (1 M KFSI in EC/DEC (1:1, v/v)). To fabricate the cell, the working and counter electrodes were first placed at the opposite inner sides of the quartz cuvette, along with the injection of the electrolyte. Then, a silicone plug was used to seal the cell and immobilize the electrodes (with part of the Cu foil located outside of the cell). The cell was connected with a NEWARE CT-4008T tester for K plating experiments. Optical microscopy (Zeiss Smartzoom 5) observation was carried out by applying a special device provided by Beijing Science Star Technology Co., Ltd. For the operando XRD electrode preparation, MSCNFs were cut into circular pieces 8 mm in diameter and directly utilized as cathodes for K deposition, with Al foil as the substrate and window for X-ray transmission. For a control experiment, an Al foil was directly used for K deposition. The Swagelok-type operando cell is composed of stainless steel (SS) positive/negative cases for mechanical support and electric conduction, a ceramic lining for insulation and space confinement, O-rings for cell sealing and a spring for electrode fixation. Inside the ceramic lining, the electrodes were fabricated in the order of cathode (MSCNFs), separator (glass fiber), anode (K foil) and SS spacer (Supplementary Fig. 27). For the K−S/K-PPB full cells, the MSCNF-K anode was coupled with a Se$_{0.05}$S$_{0.95}$@pPAN/PPB cathode, glass fiber separator and 1 M KFSI in EC/DEC (1:1, v/v) electrolyte (80 μL). The K−S and K-PPB cells were cycled within the voltage ranges of 1.0–3.0 V and 2.0–4.0 V, respectively. In situ electrochemical impedance spectroscopy (EIS, potentiostatic) was measured using an electrochemical workstation (Bio-Logic SP-300, France) with an amplitude of 50 mV and 6 data points per decade in logarithmic spacing. The frequency range was 1 MHz to 100 mHz. Before EIS measurement, an open-circuit voltage time of 10 min was applied to rest the cell. Linear sweep voltammetry (LSV) with a sweep rate of 1 mV s$^{-1}$ and a voltage range from −200 to 200 mV was used to determine the charge-transfer kinetics of the electrode surface. The exchange current density was calculated using a linear fit of Tafel plots (from 100 to 150 mV). All of the above electrochemical tests were carried out at 25 ± 2 °C without using a climatic/environmental chamber.

## Theoretical calculations

Density functional theory calculations were performed by the Cambridge serial total energy package[75] code, in which a plane wave basis set was used. The exchange and correlation interactions were modeled using the generalized gradient approximation and the Perdew–Burke–Ernzerhof[76] functional. The Vanderbilt ultrasoft pseudopotential[77] was used with a cut-off energy of 450 eV. Geometric convergence tolerances were set for a maximum force of 0.03 eV/Å, maximum energy change of 10$^{-5}$ eV/atom, maximum displacement of 0.001 Å and maximum stress of 0.5 GPa. The sampling in the Brillouin zone was set to 3 × 3 × 1 by the Monkhorst–Pack method. The adsorption energy ($E_{ads}$) was defined as $E_{ads} = E_{Surf-K} - E_K - E_{Surf}$ where $E_{Surf-K}$ is the total energy of graphite/graphite-N/graphite-N@Zn adsorbed with K, $E_K$ is the total energy of K, and $E_{surf}$ is the total energy of graphite/graphite-N/graphite-N@Zn.

## Electric field simulations

The finite element analysis simulations of the electric field were performed by COMSOL Multiphysics. Simplified 3D models of randomly arranged carbon fibers (CFs) and porous fibers (representing MSCNFs) were set up, where the fibers of each layer with an alternating direction were stacked horizontally on the $X$−$Y$ plane, and the spacing on the Z-axis between layers was fixed. The radius of the CFs was set as 1 μm based on the SEM images. Eight layers were included in the simulation domain, where the total z-directional distance was 2.5 μm. The simulation domain for MSCNFs was modified based on the geometry of the CFs. Solid fibers (thickness of 1 μm) with elliptic holes were randomly generated to mimic the porous structure of MSCNFs. The static electric field under a fixed potential drop over the z direction was simulated. The electrical conductivities of the CFs and electrolyte were 10 and 1 S cm$^{-1}$, respectively.

## Data availability

All data generated in this study are provided in the Source Data file and Supplementary Information. Source data are provided in this paper.

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

## Acknowledgements

This work was supported by the National Natural Science Foundation of China (Nos. U1966214, 51821005, 21975087, 22008082) and the Certi-ficate of China Postdoctoral Science Foundation Grant (2021M691134). We gratefully acknowledge the Analytical and Testing Center of HUST for allowing us to use its facilities.

## Author contributions

S.W.L. designed and performed the experiments and prepared the manuscript; H.L.Z., Y.L., and Z.L.H. performed the characterization experiments; L.F.P., S.P.L., and C.Y. contributed to the discussion of the experiments and the processing of experimental data; S.J.C. provided experimental resources; J.X. supervised the work, helped to prepare the manuscript and provided experimental resources; all authors con-tributed to the discussion and the manuscript preparation.

## Competing interests

The authors declare no competing interests.
