## [Peer Review File · Nature Communications]

REVIEWER COMMENTS

Reviewer #1 (Remarks to the Author):

The manuscript reports on a new host strategy for stable K-metal deposition. Overall the study shows very interesting results, with a very high capacity of the new material bearing promise for future real applications. The new findings and insights presented in the manuscript are of high interest and thus could very well be considered for publication, however only after addressing several issues which are need to be clarified to understand the full impact of this work.

i) In the introduction the authors should better introduce state of the art to set their work in perspective. The motivation for the choice of Zn as a dopant for the MSCNF, also by relating to previous work in literature.

ii) Role of formation cycles? Are the formation cycles performed before the electrochemical tests (in e.g. fig 2) or is for instance the data shown in fig 2a part of the formation cycle?

iii) In the data shown in figure 2a a very small overpotential is found for the MSCNF. However, an additional difference between the nano-fiber materials and Cu is the delay in the nucleation. Could the authors explain this? Is this related to SEI formation and if so, why does it seem to here consume much more capacity than for Cu? In relation to SEI formation it would be valuable to actually see the voltage profile and the nucleation potential in subsequent cycles.

iv) The MSCNF material is very porous (meso and micro) and has also considerable amount of voids between the fibres. While this is beneficial for depositing a large amount of K in the structure it will also require very large amount of electrolyte. The authors should comment on this fact in relation to a final energy density of the cell.

v) In cross sectional image S30 (should actually be in main text for clarity) it looks like a considerable amount of K is deposited on top of the MSCNF host (a similar impression is given by figure 3f as well). The authors should clarify this, how much is in the host and how much is on top of the host material? In

fact in figure 3b it also looks like some deposition occurs on top, but this figure is very difficult to interpret.

vi) If the majority of the K-metal is deposited in the host structure one would in fact expect a certain volume change of the electrode, since this kind of fibre structure is not very rigid. This should be confirmed by cross-sectional images with SEM or other methods.

vii) A coulombic efficiency of 97% is stated. But for long cycle life this is not really enough, one would very fast deplete the cell of K at this CE level. For the experiments reported here a huge excess of K is present in the cell. The authors should comment on how to reach values needed for real applications (>99.99%) where there is no excess K in the cell.

viii) A minor point is that how the specific capacity was calculated should be discussed in main text for clarity. Now it is discussed in SI.

Reviewer #2 (Remarks to the Author):

Dear Authors,

The manuscript is well presented, complete and contains high-quality science that will be of significance to the field of batteries. The novelty of the paper is quite high despite many small typos found along with the manuscript. The authors carefully tailored the synthesis, performed many characterizations of the MSCNFs anode materials and, finally, they evaluated the electrochemical performance in a full Sulfur cells. I found the choice of testing the material in a sulfur cell very safe and low innovative but the material has high potassiphilicity, shows great performance and, therefore, the value of the manuscript remains quite high. I recommend the publication in nature communication after addressing the following comments.

-Please, find enclosed the attached file, which contains all the typos check in the manuscript and fix them.

-Figure 3 c and d. Define this measure In-situ XRD is completely wrong. The measure is an operando XRD analysis. I kindly ask the authors to describe the electrode preparation and the operando cell setup used for the reported measurement.

-Figure 4. Please, add in the caption that the spectra are XPS spectra. It might be not obvious to some young readers.

-Etching of the MSCNFs. Could, the authors elaborate, please, on the choice of HCl? And add this explanation in the preparation section.

-As I said, I found too safe the choice of coupling the anode with a sulfur-base cathode. The manuscript shows that the anode works for conversion materials. However, does it work with intercalation type cathodes? The title of the manuscript does not mention any sulfur system, so I suppose that these new materials are enabling the use of potassium metal anodes in different kinds of systems.

Responses to reviewers' comments

Reviewer #1 (Remarks to the Author):

The manuscript reports on a new host strategy for stable K-metal deposition. Overall the study shows very interesting results, with a very high capacity of the new material bearing promise for future real applications. The new findings and insights presented in the manuscript are of high interest and thus could very well be considered for publication, however only after addressing several issues which are need to be clarified to understand the full impact of this work.

Response:

We appreciate your positive comments and recommendation on this article. We are grateful for your support on this work. And we also thank you for your specific comments below. In general, we sincerely hope that the supplemented data added and the further explanations given will eliminate your concerns.

i) In the introduction the authors should better introduce state of the art to set their work in perspective. The motivation for the choice of Zn as a dopant for the MSCNF, also by relating to previous work in literature.

Response:

To give a detailed introduction, the following information is provided:

Typically, metal-based hosts such as Cu and Al are a class of potential hosts for stable and dendrite-free composite K anode considering their commercial availability. After proper treatment to increase the potassiophilicity, these modified hosts including rGO@3D-Cu (Cu foam coated with reduced graphene oxide), Al@Al (aluminum foil coated with aluminum powders) and Cu₃Pt-Cu mesh (Cu₃Pt functionalized-Cu meshes) show remarkable improvement in stable K plating and stripping (Adv. Mater. 2020, 32, 1906735; Adv. Mater. 2020, 32, 2002908; Nano Energy 2020, 75, 104914). These metal-based hosts all demonstrate improved CE and suppressed overpotential, proving the validity of introducing potassiophilic sites to hosts. However, the intrinsic high density and relatively low space utilization hinder the application of metal hosts.

In contrast, carbon-based hosts are more favorable for K accommodation mainly because of their lightweight and electrochemical stability, which can realize higher volumetric or gravimetric energy density. HNCP/G (hollow N-doped C polyhedrons/graphene composite) and aligned carbon nanotube membrane (ACM), as pure carbon hosts, can realize prolonged symmetric cells cycling time (> 100 h) and decreased overpotential (< 0.1 V) (ACS Nano 2019, 13, 9306–9314; Adv. Energy Mater. 2019, 9, 1901427). In these cases, robust and highly conductive carbon-based hosts successfully evidence their potential for K accommodation. Consequently, an incorporation of carbon-based materials with other kinds of potassiophilic species becomes popular to deliver enhanced K plating/stripping electrochemistry. For example, PM/NiO (puffed millet/NiO) gives an enhanced K deposition volume and smaller voltage hysteresis due to the collaboration of PM host and potassiophilic NiO nanoparticles (Nano Energy 2019, 62, 367–375), PCNF@SnO₂ (SnO₂-coated conductive porous carbon nanofiber) realizes high K uptake (87%@15 mg cm⁻²) and uniform K nucleation taking advantage of its void-rich carbon nanofibers and potassiophilic SnO₂ coating (J. Mater. Chem. A 2020, 8, 5671–5678), DN-MXene/CNT (defect-rich and nitrogen-containing MXene/carbon nanotube) achieves high CE (98.6%) and prolonged cycle life on account of its titanium defects and interconnected carbon scaffolds (Adv. Mater. 2020, 32, 1906739). Beyond the above metal compounds, metal nanoparticles might also draw attention for their potential of high K affinity and ease of synthesis. Previously, Au (Nat. Energy 2016, 1, 16010; Adv. Energy Mater. 2018, 8, 1802352), Ag (Adv. Mater. 2017, 29, 1702714; Sci. Adv. 2021, 7, eabg3626) and Zn (Adv. Energy Mater. 2018, 8, 1703505; Angew. Chem. Int. Ed. 2021, 60, 8515–8520) nanoparticles have been proven to be advantageous to lead a heterogeneous seeded growth of Li on carbon hosts. These results suggest that an introduction of metal nanoparticles (Pb, Sb, Sn and Zn) good at forming a K alloy would be an efficient strategy to improve the potassiophilicity of carbon hosts.

Per request, the corresponding introduction has been given in the revised manuscript (page 4).

ii) Role of formation cycles? Are the formation cycles performed before the electrochemical tests (in e.g. fig 2) or is for instance the data shown in fig 2a part of the formation cycle?

Response:

“Formation cycles” is usually applied prior to electrochemical testing of metal anodes, it is a standard practice to have the working electrode current collectors undergo several discharge/charge cycles aiming to stabilize the SEI before plating and remove any electrochemically unstable residual impurities (Adv. Mater. 2020, 32, 2002908; Adv. Mater. 2020, 32, 1906735). In our study, formation cycles are performed for electrochemical tests relating to half cells such as CE tests (Figure 2c), operando XRD tests (Figure 3c, d) and morphology measurements of K deposition (Figure 3e, f).

We have included the above information in the revised manuscript (page 9; method section).

iii) In the data shown in figure 2a a very small overpotential is found for the MSCNF. However, an additional difference between the nano-fiber materials and Cu is the delay in the nucleation. Could the authors explain this? Is this related to SEI formation and if so, why does it seem to here consume much more capacity than for Cu? In relation to SEI formation it would be valuable to actually see the voltage profile and the nucleation potential in subsequent cycles.

Response:

From previous reports, it is a common phenomenon for carbon-based or highly porous hosts to experience a delay in the nucleation process compared to nonporous metal substrates such as Cu foils, which can be ascribed to SEI formation that consumes more electrolytes on the substrates with large surface area (Nat. Energy 2016, 1, 16010; Adv. Mater. 2020, 32, 1906739; Adv. Energy Mater. 2020, 2002654; Energy Technol. 2021, 9, 2000700).

By comparing the subsequent voltage profiles of the MSCNF||Cu cell with its 1st cycle (Figure R3), it can be observed that the trend of delay gradually weakens and

completely disappears at the 5th cycle, in consistent with its relatively low coulombic efficiency in the initial cycles. Similar trends exist in K-graphite batteries using carbonate electrolytes, where the graphite electrode exhibits a short voltage plateau near 0.7 V on the first potassiation curve and disappears in the following cycles (Adv. Funct. Mater. 2016, 26, 8103–8110; Energy Storage Mater. 2020, 24, 319–328). Besides, the nucleation overpotentials in the subsequent cycles stay at ~20 mV, which show neglect variation. Consequently, it can be speculated that the delay of K plating on MSCNFs in the nucleation process is mainly due to the SEI formation at the 1st discharge, and the porous feature of MSCNFs make them consume much more capacity than Cu foils during this process.

We have included the corresponding discussion and Figure R3 in the revised manuscript (page 9) and Supplementary Information (Supplementary Figure 10b).

Figure R3 Voltage profiles of K plating on MSCNFs at different cycles at a current density of 0.5 mA cm^{-2} .

iv) The MSCNF material is very porous (meso and micro) and has also considerable amount of voids between the fibres. While this is beneficial for depositing a large amount of K in the structure it will also require very large amount of electrolyte. The authors should comment on this fact in relation to a final energy density of the cell.

Response:

This is a very good perspective that clearly points out the critical issue that metal anodes based on a porous-host design currently confront for practical applications. It is also a problem for MSCNF. Actually, in this work, to demonstrate the advantage of our material design for outstanding cycling performance under a high K loading, we

used composite MSCNF-K anodes fully infused with K metal to fabricate full cells. As a result, the composite anode avoids the exposure of high surface area of MSCNF, which does not consume a large amount of electrolyte.

However, in pursuit of higher energy density, it indeed becomes a problem when we use a MSCNF host for an anode-free K battery or a K-metal battery with low N/P ratio (<2). In these systems, a MSCNF host would require excessive amounts of electrolyte and therefore decrease the energy density of the cell. In fact, there are several potential strategies that may reduce the requirement of electrolyte in a metal battery based on a porous anode host: (1) Decreasing the thickness of a porous host ($<50\ \mu\text{m}$) via optimized material preparation method. For example, a proper electrospinning method is capable of fabricating a freestanding film as thin as $10\ \mu\text{m}$ (Chem. Eng. J. 2021, 405, 126596), this proves the possibility of preparing a much thinner porous host that requires less electrolyte. (2) Employing a pretreatment such as controlled amounts of K deposition along with a formation process (several cycles of plating/stripping at a mild condition), partially lowering the host's surface area and suppressing extra interfacial side reactions that consume a lot of electrolyte (Adv. Mater. 2022, 34, 2109767). Hence, how to increase the energy density of a K metal battery based on porous anode hosts still remain a challenge for the practical application of K metal batteries, associative research is ongoing in our lab.

Per request, we have included the above discussion in the revised manuscript (page 19).

v) In cross sectional image S30 (should actually be in main text for clarity) it looks like a considerable amount of K is deposited on top of the MSCNF host (a similar impression is given by figure 3f as well). The authors should clarify this, how much is in the host and how much is on top of the host material? In fact in figure 3b it also looks like some deposition occurs on top, but this figure is very difficult to interpret.

Response:

We are sorry for our uninformative presentation of morphology images correlating with K deposition, which might cause misleading information. Actually,

there is a homogeneous K deposition taking place across the whole MSCNF host. To address the reviewer's concerns, we have now added some instructive marks and labels in the morphology images obtained from SEM and in-situ optical microscopy to better demonstrate the details. Specifically, from the cross-sectional images (Figure R4a), the MSCNF host retains its fibrous feature after a shallow discharge at 0.5 mA h cm^{-2} , and large amounts of voids can be observed within the host. This result gives the fact that the host is still in the initial state of K plating where the majority of the K metal deposits inside the pores and the surface of the nanofibers. Besides, some aggregated K particles form and uniformly distribute within the whole MSCNF host. On the contrary, there is no recognizable aggregates on the top of the host.

Figure R4 SEM images of the cross-section view of K deposited on MSCNFs at the capacity of (a) 0.5 mA h cm^{-2} and (b) 3 mA h cm^{-2} (with different magnification from left to right).

At 3 mA h cm^{-2} , the host still partially maintains a fibrous feature on the top (Figure 3f, D3) whereas its interior space is already filled with dense K metal (Figure R4b), which means K metal does not preferentially deposit on the top of MSCNF. Additionally, cross-sectional EDS elemental mappings of the plated MSCNFs at different discharged states has been given in Figure R5 and R6, which confirm the homogeneous distribution of K in the deposited host, especially for the bottom of MSCNFs near Cu foils.

Figure R5 (a) SEM image of the cross-section view of K deposited on MSCNFs at the capacity of 0.5 mA h cm^{-2} and corresponding elemental mapping of (b) K, (c) C and (d) Cu.

Figure R6 (a) SEM image of the cross-section view of K deposited on MSCNFs at the capacity of 3 mA h cm^{-2} and corresponding elemental mapping of (b) K, (c) C and (d) Cu.

As for the in-situ optical microscopy observation, it has to be pointed out that the MSCNF host shows a mossy-like morphology on the top throughout the whole plating process because of its loosely packed carbon nanofibers near the surface (Figure R7b, highlighted with red arrows). Due to a lack of space confinement from the vertical direction, the nanofibers on the top move freely under the driving force of electric field and ion flux, leading to an optical illusion that some deposition occurs on the top. In fact, it is difficult for us to investigate the precise position of K plating from this kind of optical observation, the result shown here only make us conclude that MSCNFs can effectively induce a homogeneous K deposition without the

formation of rod-like K dendrites and therefore achieve a dendrite-free K anode.

Figure R7 In-situ optical microscopy observation of K deposition on (a) Cu foil and (b) MSCNFs at a current density of 6 mA cm^{-2} .

We have now included the above discussion and Figure R4–7 in the revised manuscript (page 11 and 13; Figure 3b) and Supplementary Information (Supplementary Figure 32–34).

vi) *If the majority of the K-metal is deposited in the host structure one would in fact expect a certain volume change of the electrode, since this kind of fibre structure is not very rigid. This should be confirmed by cross-sectional images with SEM or other methods.*

Response:

Theoretically, the relationship between deposition capacity and the thickness of K plated on a flat substrate can be described *via* the following equation:

$$T = \frac{QM}{N_A e} \times \frac{1}{\rho} \quad (1)$$

where T is the thickness, Q is the areal capacity, M is the molar mass, N_A is the Avogadro constant, e is the electron charge, ρ is the density. Accordingly, it can be calculated that a K deposition capacity of 0.5, 3 and 5 mA h cm^{-2} corresponds to a thickness of 8.5, 50.7 and $84.4 \text{ }\mu\text{m}$, respectively.

The cross-sectional images of K deposited on Cu foils show a thickness of ~ 15 and $\sim 84 \mu\text{m}$ at the discharge capacity of 0.5 and 3 mA h cm^{-2} , respectively (Supplementary Figure 31). In contrast, the deposited MSCNF hosts show a thickness of ~ 59 and $\sim 58 \mu\text{m}$ at the discharge capacity of 0.5 and 3 mA h cm^{-2} , respectively (Figure R4). Therefore, we can conclude that: (1) due to a lack of effective guidance and space confinement, the K metal deposited on Cu foils tends to form a loose and porous film, whose thickness is much higher than the theoretical value; (2) MSCNFs provide distinct regulation for K deposition that guarantees a dense and flat metal composite film. What's more, the neglectable change on the thickness of the deposited MSCNF at different states verifies our speculation that the K plating process can be divided into three phases (Figure R8): Phase I, K metal starts to nucleate inside the pores of MSCNFs, finally fills up the pores and fully covers the carbon nanofibers ($< 0.5 \text{ mA h cm}^{-2}$); Phase II, the metal cover on the nanofibers continues to grow and fills up the voids between the carbon nanofibers, forming a dense composite film ($< 3 \text{ mA h cm}^{-2}$); Phase III, an extra amount of K metal continues to deposit on the MSCNF-K and thicken the film, leading to a volume change that follows the theoretical value.

Figure R8 Schematic illustration of K plating on MSCNF.

To confirm our speculation, the thickness of the MSCNF electrode discharged at 5 mA h cm^{-2} was measured. By subtracting the thickness of Cu foil ($10 \mu\text{m}$), the obtained electrode has a thickness of $\sim 90 \mu\text{m}$ (Figure R9a), which agrees with the theoretical value ($84.4 \mu\text{m}$). The cross-section view of the electrode also shows a relatively smooth surface and uniform distribution of carbon nanofibers (the dark spots belong to carbon nanofibers), indicating an effective confinement of high-loading K and simultaneous expansion of the host and K metal (Figure R9b). Consequently, Phase I and II do not affect the intrinsic thickness or volume of the

MSCNF host, and the volume change caused by Phase III can be deduced theoretically.

Figure R9 (a) Digital photo of thickness measurement and (b) cross-sectional SEM image of a MSCNF host deposited with a capacity of 5 mA h cm^{-2} .

We have included Figure R8 and R9 and the above discussion in the revised manuscript (page 13 and 14) and Supplementary Information (Supplementary Figure 35 and 36).

vii) A coulombic efficiency of 97% is stated. But for long cycle life this is not really enough, one would very fast deplete the cell of K at this CE level. For the experiments reported here a huge excess of K is present in the cell. The authors should comment on how to reach values need for real applications (>99.99%) where there is no excess K in the cell.

Response:

We admit that a CE of 97% is still far from satisfactory for real applications, since it can lead to $\sim 20\%$ loss of K metal after only 7 cycles under the CE of 97% ($0.97^7 = 80.80\%$) and a lack of excess K. In comparison, a high CE of 99.99% is more applicable for a cell applied in EV industry or energy storage concerning the requirement of 80% capacity retention over 2000 cycles ($0.9999^{2000} = 81.87\%$).

To achieve a high CE, a priority is to avoid the irreversible loss of active K during cycling, which means strategies should be focusing on preventing the further decomposition of electrolytes and the excessive or continuous formation of solid electrolyte interphase (SEI) that consumes large amounts of K. From previous reports on Li metal anodes, selection of a proper electrolyte plays a key role in forming a stable and robust SEI that correlates with increasing CE (Nat. Energy 2021, 6,

951–960). This strategy is found to be effective for K metal anodes. For example, Wang et al. reported that the CE of a MXene substrate can be improved from 93.1% to 98.6% when the KPF₆ salt was replaced with KFSI in the electrolyte (Adv. Mater. 2020, 32, 1906739), same trend was revealed by Ming and coworkers (ACS Energy Lett. 2020, 5, 3124–3131). This can be ascribed to the highly reactive fluorine bound to the S of sulfone in KFSI, which leads to a fluorine-rich SEI favorable for reversible K plating/stripping. Apart from the anion effect, solvents (for example, 1,3-dimethoxyethane) that possess low dielectric constant and high the lowest unoccupied molecular orbital (LUMO) levels are also important for their increased stability of the K⁺-solvent structure and lower possibility of solvent decomposition (ACS Energy Lett. 2020, 5, 3124–3131). Meanwhile, increasing the salt concentration can further aggregate solvents and anions and therefore shorten the distance between anions and K⁺, enhancing the stability of K⁺-solvent structure (ACS Energy Lett. 2020, 5, 3124–3131).

Briefly, apart from a highly potassiophilic and robust host, a high CE eligible for the practical application of K metal batteries (>99.99%) would be realized together with a rational electrolyte design that combines following features: (1) suitable anions such as FSI⁻ for F-rich SEI formation; (2) solvent with low dielectric constant and high LUMO levels; (3) high-concentration electrolytes or localized high-concentration electrolytes (aggregated solvent and anions) that provide a stable K⁺-solvent structure.

Per request, we have now included the above discussion in the revised manuscript (page 10).

viii) A minor point is that how the specific capacity was calculated should be discussed in main text for clarity. Now it is discussed in SI.

Response:

In detail, the specific capacity of the infused MSCNF-K electrode was collected by discharging the MSCNF-K||Cu coin cell at 0.02 mA cm⁻², whose value was determined based on the total mass of K metal and MSCNFs in the composite anode.

For a K–S full cell, the specific capacity is calculated based on the mass of Se and S in the cathode.

Per request, we have included the above information in the revised manuscript (method section).

Reviewer #2 (Remarks to the Author):

Dear Authors,

The manuscript is well presented, complete and contains high-quality science that will be of significance to the field of batteries. The novelty of the paper is quite high despite many small typos found along with the manuscript. The authors carefully tailored the synthesis, performed many characterizations of the MSCNFs anode materials and, finally, they evaluated the electrochemical performance in a full Sulfur cells. I found the choice of testing the material in a sulfur cell very safe and low innovative but the material has high potassiophilicity, shows great performance and, therefore, the value of the manuscript remains quite high. I recommend the publication in nature communication after addressing the following comments.

Response:

We are grateful for your support on this work. We want to express the gratitude on the reviewers' time and effort to help improve this manuscript. The listed issues are addressed point-by-point.

-Please, find enclosed the attached file, which contains all the typos check in the manuscript and fix them.

Response:

Per request, we have checked the typos carefully and corresponding revisions have been given and highlighted in yellow in the revised manuscript (page 2, 3, 4, 5, 6, 7, 8, 9, 10, 14, 19, 22 and 23).

-Figure 3 c and d. Define this measure In-situ XRD is completely wrong. The measure is an operando XRD analysis. I kindly ask the authors to describe the electrode preparation and the operando cell setup used for the reported measurement.

Response:

Per request, we have changed the description of in-situ XRD into operando XRD

in the revised manuscript accordingly (caption of Figure 3; page 11; method section).

For the electrode preparation, MSCNFs were cut into circular pieces of 8 mm diameter and directly utilized as the cathodes for K deposition. For a control experiment, an Al foil was used to act as both a non-porous substrate for K deposition and a window for X-ray transmission.

The operando cell setup is shown in Figure R10. Specifically, the Swagelok-type operando cell is composed of stainless steel (SS) positive/negative cases for mechanical support and electric conduction, a ceramic lining for insulation and space confinement, O-rings for cell sealing and a spring for electrodes' fixation. Inside the ceramic lining, the electrodes were fabricated with the order of cathode (MSCNFs), separator (glass fiber), anode (K foil) and SS spacer.

Figure R10 (a) Digital image and (b) schematic of the operando XRD cell setup employed in this study.

The above information has been included in the revised Supplementary Information (Supplementary Figure 27).

-Figure 4. Please, add in the caption that the spectra are XPS spectra. It might be not obvious to some young readers.

Response:

Per request, we have now added the description "XPS spectra" into the caption in the revised manuscript (Figure 5).

-Etching of the MSCNFs. Could, the authors elaborate, please, on the choice of HCl? And add this explanation in the preparation section.

Response:

As a common method, HCl etching is usually selected to remove acid-sensitive species or templates in certain composites without damaging their major or parent components (Angew. Chem. Int. Ed. 2019, 58, 12469–12475). Here we use HCl solution to react with Zn clusters in order to obtain Zn-free porous carbon nanofibers. The reaction is based on the following equation: $2\text{HCl} + \text{Zn} = \text{ZnCl}_2 + \text{H}_2$.

We have included the above information in the revised manuscript (method section).

-As I said, I found too safe the choice of coupling the anode with a sulfur-base cathode. The manuscript shows that the anode works for conversion materials. However, does it work with intercalation type cathodes? The title of the manuscript does not mention any sulfur system, so I suppose that these new materials are enabling the use of potassium metal anodes in different kinds of systems.

Response:

Per request, we have synthesized a typical intercalation type cathode, $\text{K}_{0.220}\text{Fe}[\text{Fe}(\text{CN})_6]_{0.805}$ (Potassium Prussian blue, PPB), to investigate the compatibility of the MSCNF-K anode.

Figure R11 XRD patterns of as-synthesized PPB powder.

The material was prepared as follows: 20 mL FeCl_3 (2 mmol) aqueous solution was added into 80 mL $\text{K}_4\text{Fe}(\text{CN})_6$ (1 mmol) aqueous solution under stirring. The dark blue precipitates formed immediately and were collected after aging for 24 h to obtain the PPB powders. The powders were thoroughly washed with deionized water and ethanol by centrifugation, and dried under vacuum at 80 °C for 12 h. PPB electrodes were fabricated by rolling the mixture of $\text{K}_{0.220}\text{Fe}[\text{Fe}(\text{CN})_6]_{0.805}$ powders, carbon

nanotubes (CNTs) and polytetrafluoroethylene (PTFE) into a thin film at a weight ratio of 7:2:1. The total mass loading of the electrodes is ~ 2 mg. The specific capacity of a K-PPB full cell is calculated based on the mass of $\text{K}_{0.220}\text{Fe}[\text{Fe}(\text{CN})_6]_{0.805}$ in the cathode.

Figure R12 Voltage profiles of (a, b) MSCNF-K||PPB and (c, d) K||PPB full-cells at (a, c) different cycles and (b, d) different specific current rates (50, 100 and 200 mA g^{-1}).

As shown in Figure R11, the X-ray diffraction (XRD) patterns of the as-synthesized PPB powder matches well with the standard pattern of $\text{Fe}_4[\text{Fe}(\text{CN})_6]_3$ (JCPDS No. 52-1907), indicating the structure of Prussian blue. K-PPB full cells were fabricated by pairing the PPB cathode with different potassium anodes MSCNF-K and bare K. The obtained MSCNF-K||PPB and K||PPB cells show identical galvanostatic charge/discharge profiles with a voltage plateau between 3.1–3.4 V (Figure R12a, c) at a specific current of 50 mA g^{-1} , in consistent with previous reports (Adv. Funct. Mater. 2017, 27, 1604307). However, the MSCNF-K||PPB cell delivers a lower voltage gap (0.26 V) than K||PPB (0.39 V) and retains much higher specific capacity at higher current rates, which is similar to the trend in K-S batteries (Figure R12b, d and R13). This result confirms the outstanding electrochemical performance

and compatibility of MSCNF-K anodes in different kinds of systems.

Figure R13. Rate performance of MSCNF-K||PPB and K||PPB full-cells.

We have now included the above experimental details and discussion in the revised Supplementary Information (Supplementary Figure 42–44) and manuscript (page 17; method section).

REVIEWERS' COMMENTS

Reviewer #1 (Remarks to the Author):

The authors have properly addressed all raised points in the first report. The manuscript can now be recommended for publication.

Reviewer #2 (Remarks to the Author):

Dear Authors,

thanks for addressing all my comments and improving the quality of the manuscript. In my opinion, the paper is now ready to be accepted for publication in Nature Comm.

Best regards.